# Benchmarking of SpCas9 variants enables deeper base editor screens of *BRCA1* and *BCL2*

Annabel K. Sangree [1,2], Audrey L. Griffith [1,2], Zsofia M. Szegletes [1], Priyanka Roy [1], Peter C. DeWeirdt[1], Mudra Hegde [1], Abby V. McGee [1], Ruth E. Hanna[1] & John G. Doench [1✉]

Numerous rationally-designed and directed-evolution variants of SpCas9 have been reported to expand the utility of CRISPR technology. Here, we assess the activity and specificity of WT-Cas9 and 10 SpCas9 variants by benchmarking their PAM preferences, on-target activity, and off-target susceptibility in cell culture assays with thousands of guides targeting endogenous genes. To enhance the coverage and thus utility of base editing screens, we demonstrate that the SpCas9-NG and SpG variants are compatible with both A > G and C > T base editors, more than tripling the number of guides and assayable residues. We demonstrate the performance of these technologies by screening for loss-of-function mutations in *BRCA1* and Venetoclax-resistant mutations in *BCL2*, identifying both known and new mutations that alter function. We anticipate that the tools and methodologies described here will facilitate the investigation of genetic variants at a finer and deeper resolution for any locus of interest.

[1] Genetic Perturbation Platform, Broad Institute of MIT and Harvard, 75 Ames St., Cambridge, MA, USA. [2]These authors contributed equally: Annabel K. Sangree, Audrey L. Griffith. ✉email: jdoench@broadinstitute.org

Coupling CRISPR technology to highly parallel methods to write and read DNA[1–3], as well as viral technologies that enable delivery to nearly any cell type of interest, has enabled pooled genetic screens across diverse models, assays, and fields[4]. However, off-target activity may occur due to recognition of an unanticipated protospacer adjacent motif (PAM) or tolerance of mispairing between the guide and DNA[5–7]. Several high-fidelity versions of SpCas9, including SpCas9-HF1[8], HypaCas9[9], and eSpCas9-1.1[10], developed by rational design, and HiFi Cas9[11] and evoCas9[12], identified by randomized screening in bacteria and yeast, have been shown to attenuate off-target cutting without substantial loss of on-target activity. Additionally, to expand the targeting-scope of SpCas9, several groups have created PAM-flexible variants. SpCas9-VQR and SpCas9-VRER, both the result of directed evolution in bacteria, are characterized as recognizing NGA or NGCG PAMs, respectively[13]. SpCas9-NG was rationally engineered to recognize NG PAMs[14] while xCas9-3.7, identified by phage-assisted continuous evolution (PACE), has been reported to recognize NG, NNG, GAA, GAT, and CAA PAMs[15], although its performance has been questioned[16,17]. More recently, SpG was developed to recognize NG PAMs, while SpRY has been characterized as essentially PAM-less[18]. Together, these variants enable targeting of genomic loci previously inaccessible by wild-type (WT) SpCas9.

Several groups have compared these variants. One study profiled HypaCas9, eSpCas9-1.1, and Cas9-HF1 using tagmentation-based tag integration site sequencing (TTISS)[19], and observed a trade-off between specificity and activity. Notably, this study included only sgRNAs containing 5′ matched guanines, as high-fidelity variants do not tolerate a mismatched 5′ guanine[8,9,20], which is prepended to enhance transcription from the U6 promoter[21]. Another study employed transient transfection and high throughput sequencing to analyze the editing efficiency, specificity, and PAM compatibility of high-fidelity and PAM-flexible variants at multiple target sites[22]. Additionally, Legut et al.[16] and Kim et al.[23] assayed xCas9-3.7 and Cas9-NG alongside WT-Cas9 in pooled screens. The former used a flow cytometry-based assay targeting three cell-surface genes with guides using all possible 3 nucleotide PAMs, the latter used a library-on-library approach to profile PAM preference. Both found that Cas9-NG is more active at NGH sites than WT-Cas9, but that PAM flexibility comes at the cost of reduced efficacy[16].

Previously, we and others have demonstrated the utility of base editor screens to introduce nucleotide variants at endogenous loci[24,25], identifying loss-of-function (LOF) mutations in clinically-relevant genes, as well as variants that modify small molecule-target interactions. We created libraries to characterize new SpCas9 variants, identified Cas9-NG and SpG as PAM-flexible variants that perform well, and developed these enzymes for base editing applications. We additionally assessed ABE8e, an A > G editor[26], for use in pooled screens. We uncover LOF mutations in *BRCA1*, and demonstrate the utility of this approach for mapping drug-target interactions by resistance screens with Venetoclax and *BCL2*.

## Results

For each Cas9 variant we systematically investigated on-target activity: cutting efficiency at intended sequences; and off-target activity: cleavage at unintended sites. All variants were included in a preliminary PAM-mapping assay, except HiFi-Cas9. The strongest high-fidelity and PAM-flexible variants were further assessed for off-target promiscuity, at which point HiFi-Cas9 was added to the panel (Fig. 1a). We established A375 cells stably expressing these Cas9 variants, and confirmed comparable expression levels via flow cytometry (Supplementary Fig. 1a, b).

To quantify performance among SpCas9 proteins, we designed a PAM-mapping library that reports on both PAM preferences and on-target efficacy, based on a variant's ability to distinguish between essential[27] and nonessential genes[28]. The library contains 70–100 sgRNAs per four nucleotide PAM, including all 256 possible PAMs (the canonical SpCas9 PAM is NGGN). Each of these sets includes three sgRNA 5′-types that differ in length and presence of a matched 5′ guanine: G19, a 20mer with a matched guanine; G20, a 21mer with a matched prepended guanine; and g20, a 21mer with a mismatched prepended guanine (Fig. 1b). This library (18,768 guides) was screened in duplicate at >500x coverage for three weeks (Fig. 1c). At the end of the screen we collected cells, isolated genomic DNA, retrieved the library by PCR, and sequenced to determine guide abundance

We first calculated the log2-fold-change (LFC) relative to the plasmid DNA (pDNA) (Supplementary Data 1). All variants showed good reproducibility (Pearson correlation 0.54–0.87) except Cas9-VRER (0.25), which had few active guides (Fig. 1d), explaining the poor reproducibility (Supplementary Fig. 2a). We quantified the fraction of guides targeting essential genes that were more depleted than the 5th percentile of guides targeting nonessential genes and non-targeting controls for every PAM. For WT-Cas9, this confirmed the preference for an NGGN PAM, as 95.3% of guides were active by this lenient definition (Fig. 1e), with low but detectable activity at NAGN (18.6%), NGAN (6.1%) and NCGN (4.7%) (Supplementary Fig. 2a). We examined the data via an alternative metric, calculating average recall at 95% precision for each variant, designating guides targeting essential genes as true positives and those targeting nonessential genes as false positives. These metrics produce concordant results (Pearson's $r = 0.98$ for WT-Cas9) (Supplementary Fig. 2b), with an average recall of 90.3% for an NGGN PAM, 18.7% at NAGN, 5.2% at NGAN, and 4.3% at NCGN PAMs.

As expected, the high-fidelity variants were only active at NGGN PAMs (Supplementary Fig. 2a), so we included only these PAMs in subsequent analyses. We calculated precision-recall curves, including guides of all 5′-types. At 95% precision, WT-Cas9 performed best (90% recall), followed by eCas9-1.1 (40%), HypaCas9 (27%), Cas9-HF1 (25%), and evoCas9 (4%) (Fig. 1f). When discretized by 5′-type, we observed a pronounced preference for G19 guides with all high-fidelity variants; this preference was marginal for WT-Cas9 (Fig. 1g). Considering only the G19 guides, WT-Cas9 again performed best (94%), followed by eSpCas9-1.1 (90%), Cas9-HF1 (76%), HypaCas9 (74%), and evoCas9 (35%), consistent with a recent study using reporter construct screens[17]. That an extra 5′G, whether paired or not, greatly diminishes activity with these variants is consistent with prior reports[19–21]. Importantly, this constraint reduces the number of potential sgRNAs four-fold. We summarized PAM activity into active (guides with fraction active >0.7) and intermediate (0.3–0.7) bins for 21mers (G20/g20) and 20mers (G19) for these variants at all PAMs (Supplementary Fig. 2c).

**Off-target activity of high-fidelity variants**. We next compared the off-target tolerance of select high-fidelity variants to WT-Cas9. To systematically assess off-targets due to mismatches with the sgRNA, we collated a set of 21 sgRNAs that were active with every high-fidelity variant in the PAM-mapping assay (all G19 guides), and included all possible single ($n = 1197$) and double mismatches ($n = 32,319$) in the sgRNA sequence, as well as 1000 non-targeting controls, for a library of 34,537 guides (Fig. 2a). We performed screens in duplicate in A375 cells stably expressing WT-Cas9; eCas9-1.1, the best-performing variant in the PAM-mapping library; and HiFi Cas9, which we had not previously assessed[11]. Replicates were well correlated (Pearson's

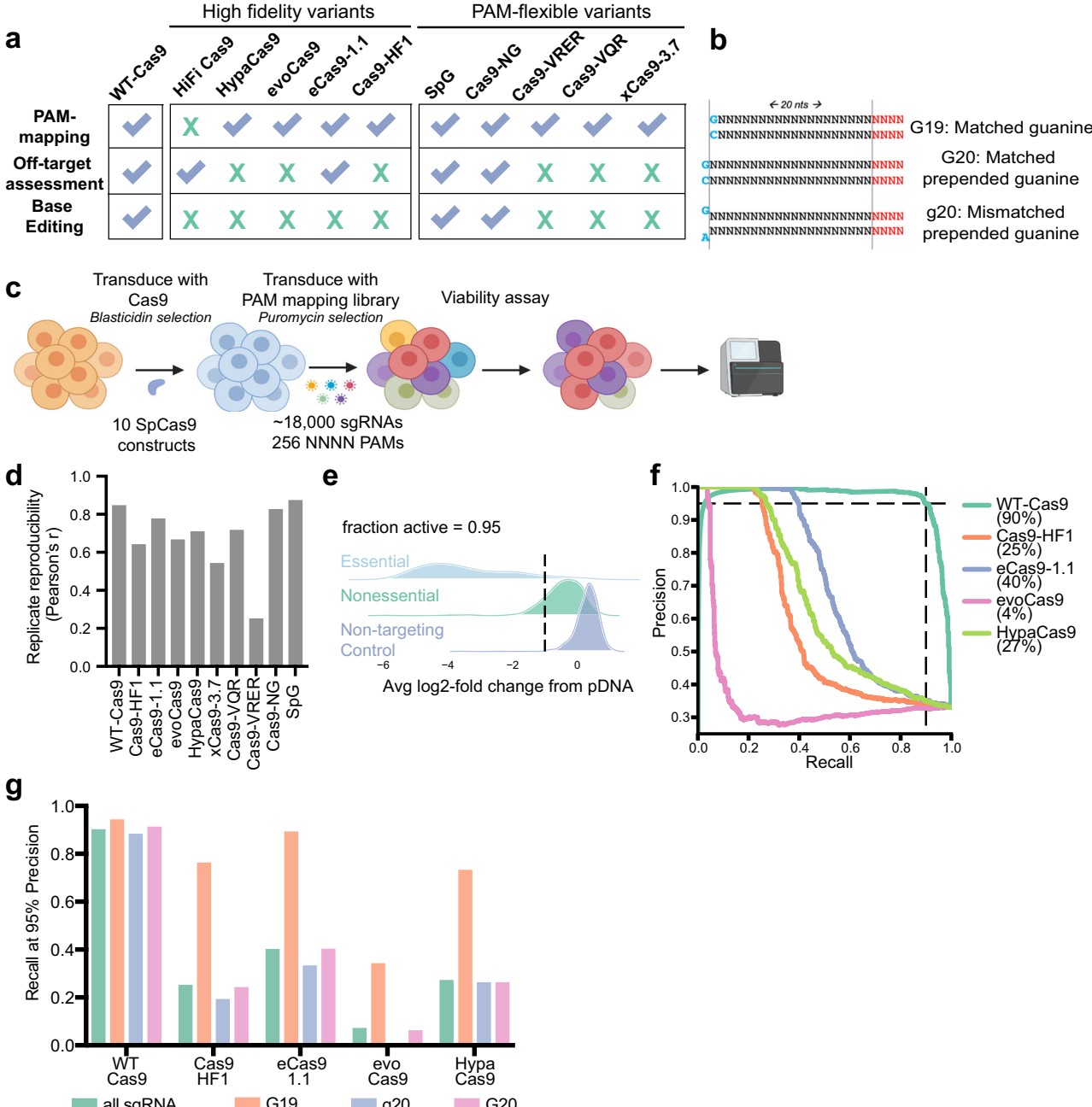

**Fig. 1 Establishment of a benchmarking assay for Cas9 activity. a** Summary of assays discussed in this manuscript and the Cas9 enzymes studied in each assay. HiFi-Cas9 was added at the off-target characterization stage, as our initial clone contained a mutation. **b** Schematic of the three 5′-types screened. Location of the PAM sequence is indicated in red. **c** Schematic of PAM-mapping screens. **d** Replicate correlation (Pearson's r), calculated from $n = 2$ experimental replicates for each variant screened. **e** Example fraction active calculation for WT-Cas9 at NGGN PAMs. **f** Precision-recall curves for WT-Cas9 and high-fidelity variants profiled with the PAM-mapping library. Guides of all 5′-types are included in this calculation. Dashed lines designate the recall at 95% precision for WT-Cas9. **g** Recall values at 95% precision for WT-Cas9 and high-fidelity variants profiled with the PAM-mapping library (NGGN PAMs only), discretized by 5′-type. Source data are provided as a Source Data file.

$r = 0.90$–$0.93$, Supplementary Data 2) and we determined the LFC of perfect match, single mismatch, double mismatch, or non-targeting control guides (Fig. 2b). To quantitate off-target activity, we calculated the ROC-AUC measuring the separation between perfectly matched guides (true positives) and mismatched guides (true negatives) (Fig. 2c). We then calculated the probability of being active for each mismatch type and position to generate a cutting frequency determination (CFD) matrix for each variant, as done previously with SpCas9 and AsCas12a[6,29] (Fig. 2d, Supplementary Table 2). We used a logistic regression model to

transform LFCs to a probability of being active, defining perfect match sgRNAs as positive controls and non-targeting sgRNAs as negative controls.

We previously generated a CFD matrix for WT-Cas9 using guides mismatched to CD33[6], and these new results were moderately consistent (Pearson's $r = 0.61$, Supplementary Fig. 3a), despite several experimental differences including 5′-type (G19 only in the present study, no requirement previously), number of genes assayed (14 vs. 1) and readout (viability vs. flow cytometry). Here, we observed a higher tolerance for mismatches at the PAM-distal

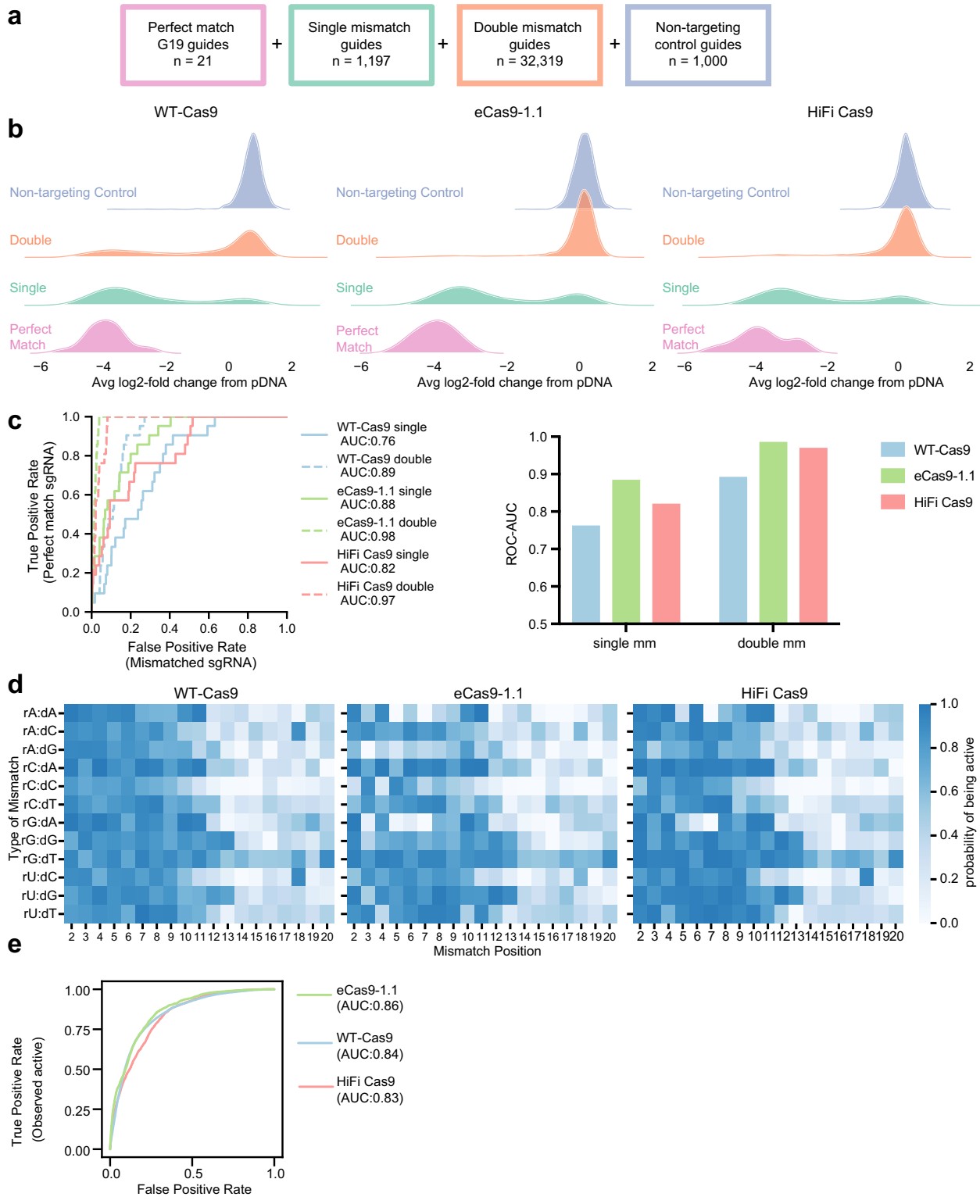

**Fig. 2 Off-target profiles of high-fidelity variants. a** Schematic depicting off-target library construction and guide selection. **b** Ridge plots showing activity of guides in the library with zero, one or two mismatches. **c** ROC plots for each enzyme screened with single (solid lines) and double mismatched sgRNAs (dashed lines). AUC is reported in the graph legend. Data are also summarized in a barplot. **d** CFD matrices for each enzyme, numbered such that 2 is the second nucleotide in the guide. Note that mismatches start at position 2, because the first position of the guide is always fixed as a G. **e** ROC plot depicting ability to predict activity at double mismatches using single mismatch data. True positives are guides that were observed to be active, false positives are guides that were not active in the screen. Source data are provided as a Source Data file.

end of the guide with all three enzymes, as well as for rG:dT mismatches (Fig. 2d), two trends observed previously in SpCas9 with other techniques and which have also been seen with Cas12a enzymes[5,6,29]. We also found that both high-fidelity variants show greater discrimination for rG:dA and rA:dA mismatches than WT-Cas9 (Fig. 2d, Supplementary Fig. 3b), whereas other mismatches, such as rC:dA and rU:dG, are still substrates for cleavage (Fig. 2d, Supplementary Fig. 3b). Using the product of the activities of each individual mismatch in the CFD matrix, we predicted the activity of double-mismatch guides, an approach that has been used to identify problematic off-target sites for SpCas9[29–31]. To evaluate our predictions, we generated ROC curves (Fig. 2e) and saw good discrimination between the two sets (AUC = 0.83–0.86) for all three enzymes, validating this approach for guide design.

**On-target activity of PAM-flexible variants**. Returning to the PAM-mapping screens, we analyzed the activity of variants that recognize alternative PAMs (Supplementary Fig. 2a). For Cas9-VQR, we observed excellent activity with all 4 NGAG PAMs (fraction active > 0.9), and poor to intermediate activity at the remainder of the NGAN PAMs (0.11–0.67) (Fig. 3a, b, Supplementary Fig. 2a). The Cas9-VRER variant, characterized to target NGCG PAMs, showed intermediate activity with GGCG (0.36) and poor activity with HGCG (0.19–0.28).

For xCas9-3.7, two PAMs showed high activity (CGGC and TGGC), with 9 additional NGGN PAMs showing intermediate activity (fraction active 0.35–0.66), and the remaining 5 NGGN showing low activity (0.16–0.29). We identified 5 additional PAMs with intermediate activity (4 NGTN, 1 NGAN) for a total of 14 intermediate PAMs (Fig. 3a, b). Legut et al. recently characterized xCas9-3.7 at all possible 64 NNN PAMs[16]. We z-scored sgRNAs targeting the coding regions of CD45 and CD55 used in their assay and observed concordance with essential sgRNAs from the present study at all PAM sites (Pearson's r = 0.79), with the majority of sgRNAs centered around 0, and activity only at NGG PAMs (Fig. 3c). Further, Kim et al.[23] performed a similar PAM classification study, measuring indel frequencies at 4 and 5 nucleotide PAMs. We compared our fraction active metric against their indel frequency using WT-Cas9 and observed good concordance (Pearson's r = 0.95, Supplementary Fig. 4a). For xCas9-3.7, we observed a similar trend (Pearson's r = 0.90), with the vast majority of PAMs centered around 0, and the strongest activity at NGGN sites, with modest activity at some NGHN sites (Supplementary Fig. 4b).

In contrast to the poor activity of xCas9-3.7, we identified 18 active and 43 intermediate PAMs with Cas9-NG[14], including high activity at NGTG and NGAG PAMs, but diminished activity at NGAC and NGCC PAMs (Fig. 3a, b), consistent with prior results[14,16,23]. We observed a similarly strong correlation between Cas9-NG in our assay and the results from Legut et al. (Pearson's r = 0.78, Fig. 3c) and Kim et al. (Pearson's r = 0.84, Supplementary Fig. 4c). Although xCas9-3.7 and Cas9-NG were both described as recognizing NG PAMs, they show little correlation (Pearson's r = 0.24, Supplementary Fig. 4d); we note that Legut et al. observed more concordance (Pearson's r = 0.72, Supplementary Fig. 4e).

Finally, we identified 24 active PAMs with SpG, all NGNN, consistent with initial characterization[18] (Fig. 3a). 41 additional PAMs showed intermediate activity, 39 of which were NGNN and the remaining 2 NANN (Fig. 3b). We next compared Cas9-NG with SpG, and observed concordance across guides (Pearson's r = 0.79 with all sgRNAs; r = 0.82 when filtered for NG PAMs) (Supplementary Fig. 4f, g), and PAMs (Pearson's r = 0.9) (Fig. 3d). We found that some NGNN PAMs were more active

with SpG than with Cas9-NG, while Cas9-NG had more activity at NANN PAMs (Fig. 3d, Supplementary Fig. 2a).

To understand if these variants had any 5′-type requirement, we calculated ROC-AUCs for each, using only active PAMs for each enzyme, and designating guides targeting essential genes as true positives and nonessential genes as true negatives (Fig. 3e). We found that none of these PAM-flexible variants demonstrated a marked preference, which is attractive for modalities like base editing.

**Off-target profiles of Cas9-NG and SpG**. To characterize the tolerance of Cas9-NG and SpG for guide-target mismatches, we selected 300 active, perfect-match sgRNAs from the PAM-mapping screens, maintaining a balance across different PAMs. We included all possible single mismatches (n = 17,775), a random subset of double mismatches (n = 60,000), and 1000 non-targeting controls, resulting in a library of 79,075 guides, including all three 5′-types (Fig. 4a). We screened this library in duplicate in A375 cells stably expressing Cas9-NG or SpG, and replicates were well-correlated (Pearson's r = 0.81 Cas9-NG; r = 0.76 SpG; r = 0.72 Cas9-NG vs SpG, Supplementary Fig. 5a, Supplementary Data 3).

We examined LFCs of perfect match, single mismatch, double mismatch, or non-targeting control guides, considering every guide included in the library (Supplementary Fig. 5b). To ensure sensitivity to mismatched guides that maintain activity, we selected 149 of the perfect match guides with the highest effect size for subsequent analyses (Fig. 4b). We applied the same framework of assessing off-target activity by calculating the ROC-AUC, comparing mismatched guides to perfect matches (Fig. 4c). We observed separation between perfect matches and single mismatches (Cas9-NG AUC = 0.86; SpG = 0.85) and excellent differentiation between perfect matches and double mismatches (Cas9-NG = 0.97; SpG = 0.97).

We then calculated the probability of being active for each enzyme with each mismatch type and position using all 5′-types to generate a CFD matrix (Fig. 4d). We also calculated matrices separated by 5′-type and compared the probabilities of being active across guide types within each enzyme (Supplementary Fig. 5c, d). For both enzymes we found that g20 guides are the least prone to off-target cutting, followed by G20, and G19 guides. We compared the probability of being active for Cas9-NG and SpG by mismatch type using all 5′-types and observed excellent concordance (Pearson's r = 0.97) (Fig. 4e). Cas9-NG and SpG have 7 and 6 total mutations, respectively, 4 of which are at the same residues, and one of which is the identical substitution (T1337R, Supplementary Fig. 5e). Thus, it is unsurprising that these variants behave so similarly.

**Base editing with PAM-flexible variants**. A major appeal of PAM-flexible variants is their potential for use in base editor screens, as the location of the perturbation is crucial for introducing the precise desired edit. While we have previously demonstrated the utility of C > T base editors (CBEs) in pooled screens, base editors capable of altering other nucleotides, such as A > G base editors (ABEs), would further expand the utility of such screens.

We benchmarked ABE7.10[32] and the newer ABE8e[26] and ABE8.17[33] in a small-scale assay using a reporter construct containing EGFP and two sgRNAs targeting EGFP delivered via lentivirus to MELJUSO cells (Supplementary Fig. 5a). After sequencing the target site, we quantitated the nucleotide percentage at each editable A (A5 and A8 with EGFP sg1 and A4 and A9 with EGFP sg2) using EditR[34]. We observed the most efficient editing with ABE8e (Supplementary Fig. 6b), so we

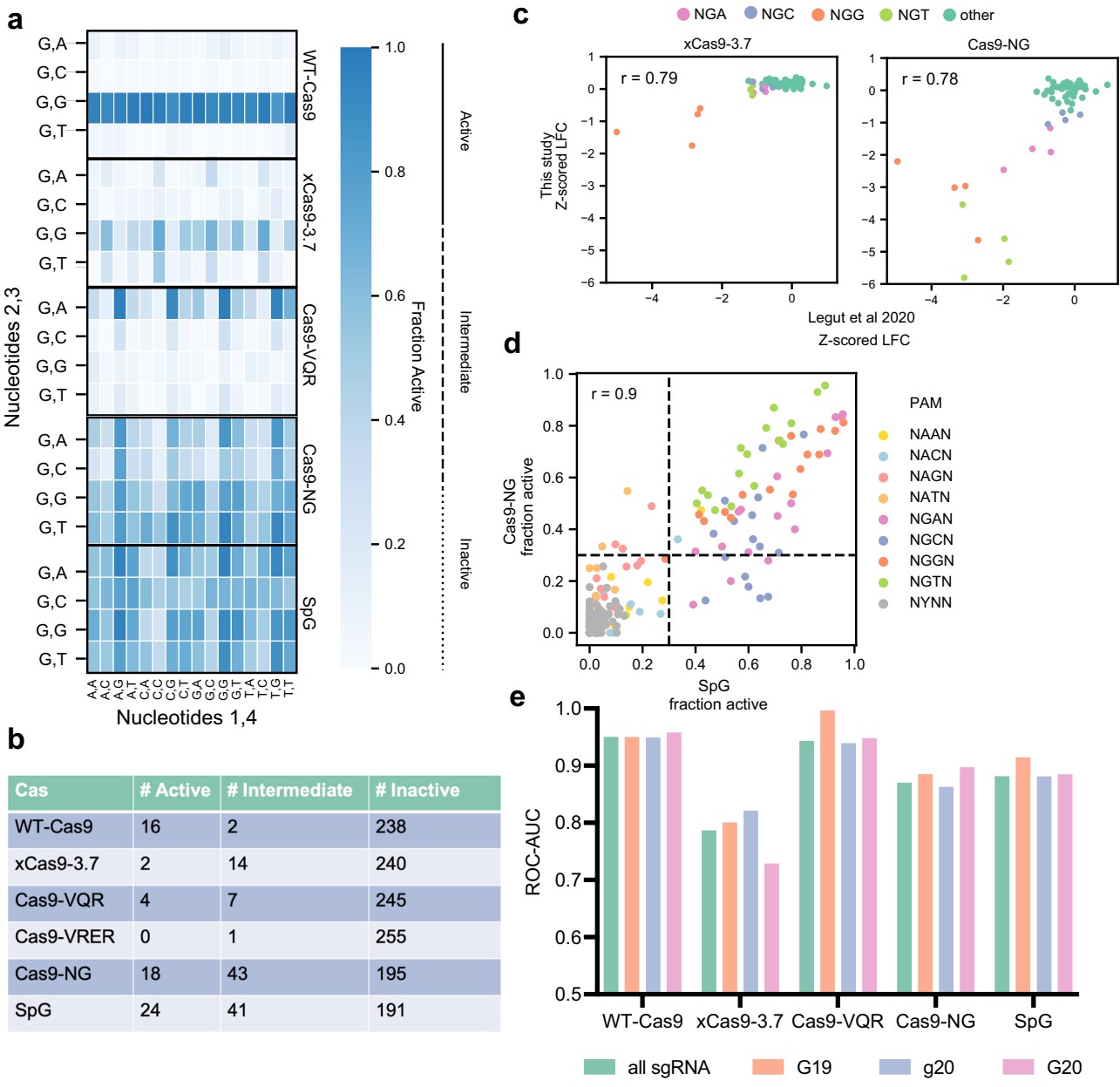

**Fig. 3 Benchmarking PAM-flexible variants. a** Heatmap of fraction active values at all NGNN PAMs. Nucleotides 1 and 4 are along the x-axis, nucleotides 2 and 3 along the y-axis. **b** Table of fraction active values for each PAM-flexible variant binned by activity bin. **c** Comparison of xCas9-3.7 (left) and Cas9-NG (right) to Legut et al. 2020. Points are colored by PAM. PAM-mapping z-scored LFC values on the y-axis refer to data in the present study. **d** Comparison of Cas9-NG and SpG fraction active values. Points are colored by PAM. Dashed lines at 0.3 indicate the cutoff for intermediate PAMs. **e** ROC-AUC values by 5′-type for each PAM-flexible variant. True positives are guides targeting essential genes, false positives are guides targeting nonessential genes. Only active PAMs are considered in this analysis. Source data are provided as a Source Data file.

selected it for further study. We next generated a Cas9-NG version of ABE8e and tested it in the same assay. Editing levels were lower compared to WT-Cas9, but still as high as 56% (Supplementary Fig. 6b).

**Base editing of *BRCA1*.** To understand the current scope of base editing screens, we used the DNA-damage repair gene *BRCA1* as an example target. For WT-Cas9, we identified 455 unique residues (24.2% of the protein) in the longest *BRCA1* isoform which could be targeted to introduce a missense or nonsense mutation (Fig. 5a). With Cas9-NG, 1342 targetable residues (71.2%) were identified considering the PAMs characterized above as active or

intermediate. Likewise, with SpG, 75.3% of the protein can be modified with at least one mutation.

Using BE3.9max and ABE8e (Supplementary Fig. 6c), we designed two base editor versions of each PAM-flexible variant: NG-CBE, SpG-CBE, NG-ABE, and SpG-ABE. To test these 4 Cas-BE variant pairings in a screen, we designed a library tiling across *BRCA1*, containing all possible guides targeting the gene, irrespective of PAM (n = 11,524), including 30 guides targeting splice sites of essential genes[27], 75 intergenic-targeting guides, and 75 non-targeting guides. By including all PAMs we hoped to further reinforce the PAM preferences of each variant, while also future-proofing this library for use with emerging Cas9 variants, such as SpRY[18]. We screened these pairings in 2 cell lines, HAP1

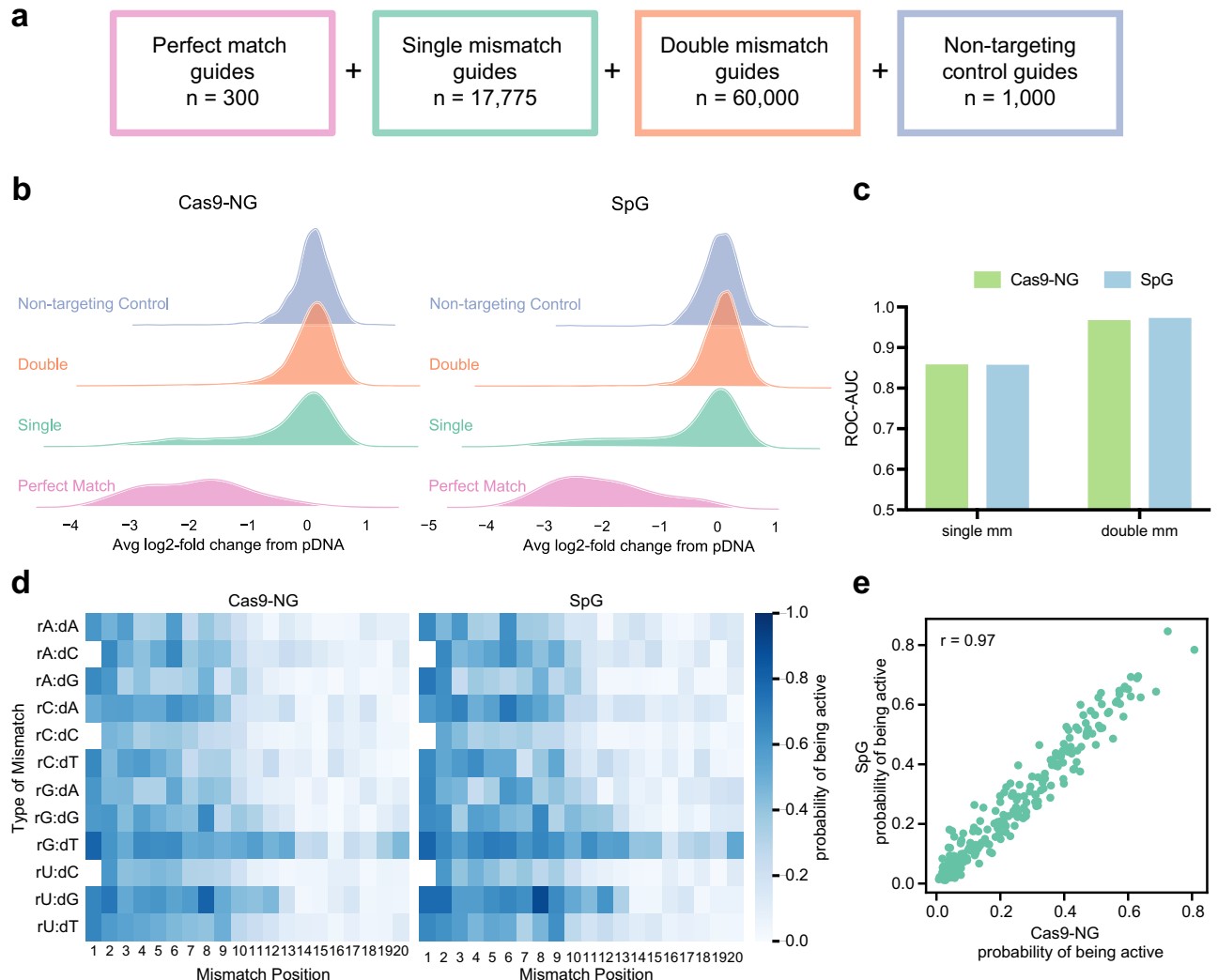

**Fig. 4 Off-target profiles of Cas9-NG and SpG. a** Schematic depicting off-target library construction and guide selection. **b** Ridge plots showing activity of filtered guides with zero, one or two mismatches. **c** ROC-AUC values at single and double mismatches for Cas9-NG and SpG. **d** CFD matrix for Cas9-NG and SpG. Note that there are no wildtype g20 and G20 guides with a G in the first position, so the rN:dC squares are blank. **e** Scatter plot showing probability of being active for each single mismatch position/type for Cas9-NG and SpG (n = 237). Pearson correlation is noted in the top left. Source data are provided as a Source Data file.

and MELJUSO at high coverage (>2000 cells per sgRNA) for 21 days (Fig. 5b). Since *BRCA1* is essential in near-haploid HAP1 cells[35], we conducted a negative selection (dropout) assay in this cell line. We treated MELJUSO cells with 1 μM of cisplatin[25] to enhance selective pressure for *BRCA1* LOF alleles. Previously, we had screened *BRCA1* with a tiling library containing only NGG PAMs using a WT-CBE[25]; we re-screened this library with WT-ABE as well.

After calculating LFCs relative to pDNA, we found that replicates were well-correlated (Pearson's *r* = 0.77–0.99 CBE screens, Supplementary Data 5; 0.77–0.95 ABE, Supplementary Data 6), and thus we averaged the data across the cell lines. We examined the distribution of positive (essential splice sites) and negative (non-targeting and intergenic) controls and found that negative controls were centered around 0, while positive controls were depleted (Supplementary Fig. 6d), confirming base editing activity. To understand our ability to assay *BRCA1* itself, we examined the separation of guides predicted to introduce nonsense or splice mutations (positive controls) and silent or no edits (negative controls) and calculated the AUC for each base editor and cell line (Supplementary Fig. 6e, f). We observed the

best performance with the WT base editors and slightly higher performance in HAP1, consistent with our original benchmarking[25]. In every condition, we observed clear separation between control groups, confirming that we were able to assay the *BRCA1* gene effectively. Next, we compared the NG and SpG base editors to WT, filtering on guides with NGGN PAMs, and observed good concordance (Fig. 5c, d), although some guides showed less activity than with WT-CBE. We speculate that this relates to the overall decreased activity with NG and SpG compared to WT at some NGGN PAMs, which may be especially important for CBE, as continued localization of the UGI domain is necessary for proper base editing. Finally, guides in the library behaved largely similarly when paired with ABE or CBE (Pearson's *r* = 0.73 SpG; *r* = 0.71 NG, Supplementary Fig. 6g), with some clear outliers.

We examined guides introducing coding changes along the length of *BRCA1* (Fig. 5e, f) and observed strong depletion of those targeting the RING and BRCT domains, consistent with our previous findings[25] and the clinical importance of these regions. However, it is inappropriate to draw conclusions about specific casual mutations from the behavior of sgRNAs solely from

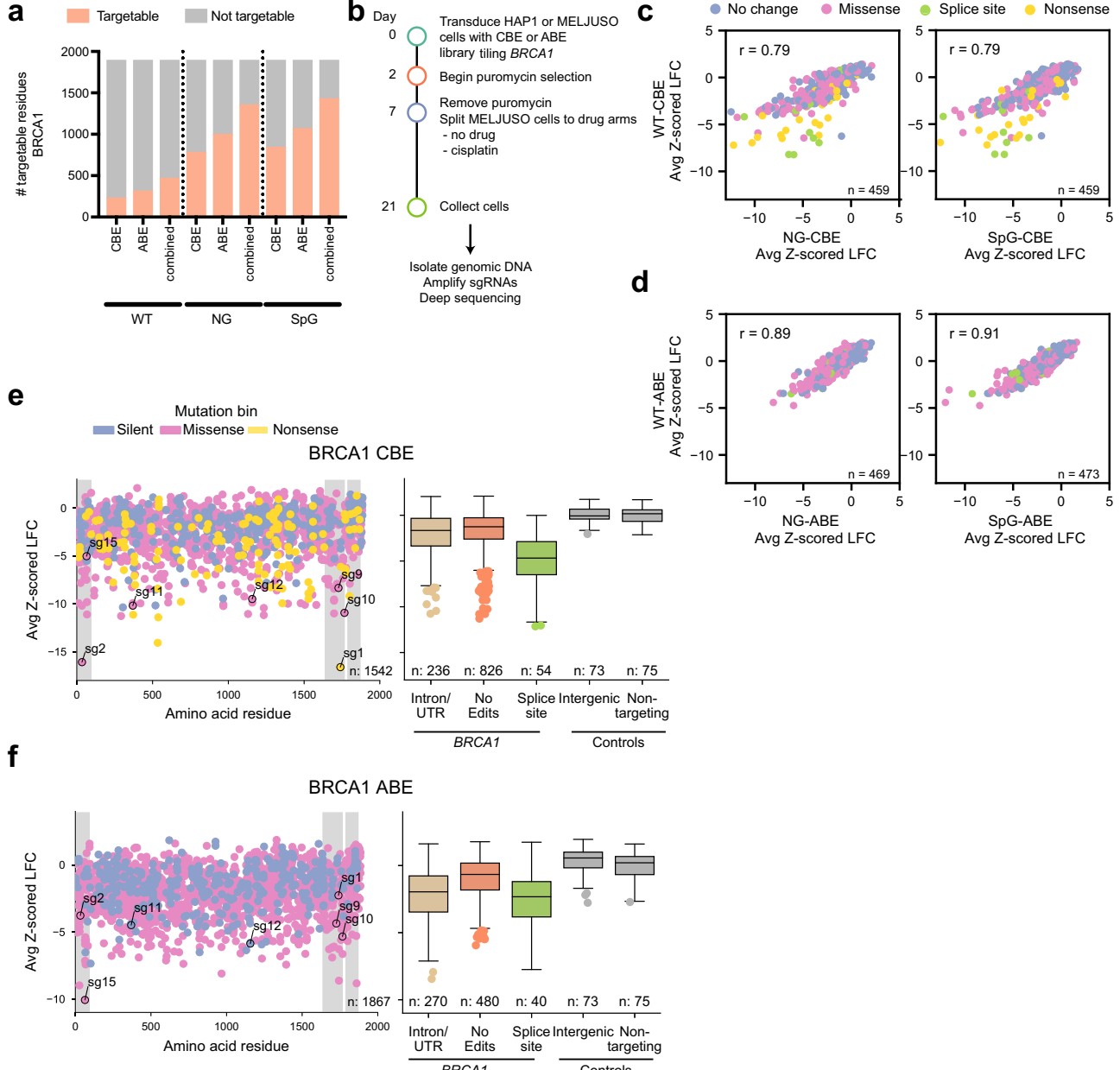

**Fig. 5 Tiling *BRCA1* with variant base editors. a** Number of targetable residues in *BRCA1* using the base editors paired with the library described in this study. **b** Timeline by which tiling screens were conducted. **c** Comparison of NG and SpG-CBEs to WT-CBE with shared guides predicted to introduce no change (silent or no edits), splice site, nonsense, or missense mutations with an NGGN PAM. Pearson's *r* is reported for each comparison. **d** Same as (**c**) but for ABEs. **e**, **f** Average performance of sgRNAs (averaged Cas9-NG and SpG screens) targeting *BRCA1*, colored according to the predicted mutation bin, for CBE and ABE screens. The first grey shaded region spans the RING domain, and the following two indicate the BRCT repeats. Boxes show the quartiles (Q1 and Q3) as minima and maxima and the center represents the mean; whiskers show 1.5 times the interquartile range (Q1-1.5*IQR and Q3 + 1.5*IQR). Source data are provided as a Source Data file.

primary screening results, as sgRNAs might deplete due to out-of-window editing, unintended edits, indels, or off-target effects[25].

***BRCA1* Validation**. We selected 18 guides (sg1-18) for validation experiments based on the magnitude of z-scores in the primary screen, regardless of PAM activity. We cloned individual sgRNAs into either ABE or CBE vectors, transduced cells, and collected samples one (early time point), two, and three weeks post-transduction. We then PCR amplified the edited locus using custom primers and deep-sequenced the edited loci to identify the causal mutations (Supplementary Fig. 7a).

At the early time point, we observed a wide range of editing efficiency (C > T, 0.04–60.1%; A > G, 0.2–59.1%). In all cases with <1% editing, the sgRNA utilized an inactive PAM. Samples with an intermediate or active PAM averaged 41.7% C > T editing and 37.4% A > G editing in the predicted edit window of 4−8 nucleotides, with lower but detectable levels outside of the window (Supplementary Fig. 7b), consistent with previous observations[25,32,36]. Next, we examined the reproducibility of percent change with WT alleles, comparing the percentage of reads in the late versus early samples (Supplementary Fig. 7c). We found that >10% enrichment of the WT allele was reproducible across replicates, and thus considered a guide to validate if the

WT allele enriched >10% from early to late samples, indicating depletion of edited alleles. By this criterion, 4/7 guides with an active or intermediate PAM that depleted in the primary screen with CBE validated, and 2/5 guides validated with ABE.

For a number of guides, we conducted validation studies with both CBE and ABE. When screened with CBE, sg2 generates a P34F mutant, which depletes from 48.0% abundance on day 8 to 11.2% on day 21 (Fig. 6a, b). Although the P34F mutation has not been documented in the ClinVar database, several publications suggest that residue 34 plays a critical role. First, it is in the RING domain, which forms a heterodimer with the RING domain of BARD1 and is necessary for E3 ubiquitin ligase activity[37,38]. Additionally, of the 6 possible missense mutations introduced at this residue via Saturation Genome Editing (SGE), 5 scored as LOF, and 1 as intermediate (P34F requires mutating 2 nucleotides, so was not included)[35]. To gain structural insight into this LOF phenotype, we visualized these residues on the crystal structure of the RING domains of BRCA1 and BARD1 (PDB IJM7). $Zn^{2+}$ atoms stabilize the structure within the RING finger and are maintained by two binding loops, Site I and Site II[38]. P34 falls between Sites I and II on BRCA1, and if mutated to F34, comes into close proximity to C66 on BARD1, part of Site II on BARD1 (Fig. 6c). While further experimentation is required to understand the exact mechanism, it seems likely that the P34F substitution destabilizes the interaction between BRCA1 and BARD1. We also examined sg2 with ABE and observed an average of 54.3% editing at positions 4 and 5, resulting in an E33G mutant. This allele remained constant across timepoints analyzed (57.9% day 8; 54.6% day 21), indicating that, in contrast to the P34F mutation, the E33G mutation is not LOF (Fig. 6d, e). Indeed, when profiled by SGE, it was classified as intermediate[35]. Further, this residue does not come in close contact with the $Zn^{2+}$ atoms or their binding loops (Fig. 6f).

We also validated sg10 with both base editors. With CBE, we observed the predicted H1767Y mutation, as well as a second Q1768X mutation caused by out-of-window editing (26.7% C > T editing at C9). Alleles with the single H1767Y mutation depleted from an average of 35.9 on day 8 to 20.8% on day 21 (Fig. 6g) while the WT allele enriched from 35.2 to 64.4%. Alleles containing only the H1767Y mutation depleted, indicating this mutation is likely sufficient for LOF (Supplementary Fig. 7d). These results are concordant with the SGE data, as both H1767Y and Q1768X individually score as LOF[35]. With ABE, sg10 introduces either an N1766S mutation (47.2% A > G conversion, A4) or N1766G mutation (18.7% A > G conversion, A3; 47.2% A > G conversion, A4) and an H1767R mutation (59.1% A > G conversion, A7), resulting in a LOF phenotype, while the WT allele enriches from 42.1% to 58.7% (Fig. 6g, Supplementary Fig. 7e). Given that the H1767R mutation occurred alone and did not deplete substantially (10.5% day 8; 9.8% day 21), it is likely that the mutants at position 1766 cause the LOF phenotype. Notably, Findlay et al. classified H1767R as functional, which is concordant with our observation; however, they also found that every missense mutation introduced at N1766 is functional, including N1766S[35]. We did not capture any alleles with a mutation only at this position, so cannot make definitive conclusions about the role of N1766S or N1766G. It remains possible that the observed LOF arises from a combinatorial effect of both mutations.

Additionally, screens with sg9 and CBE introduced a mutation in the BRCT phosphopeptide binding motif (G1727K), a conserved motif in several DNA damage repair proteins, that allows association with proteins phosphorylated by ATM (Supplementary Fig. 7f, g)[37]. Although this mutation has not been documented in ClinVar, G1727R and G1727E mutations are pathogenic, and G1727V is categorized as LOF by SGE[35],

indicating that substitutions are not easily tolerated at this position. We also screened sg15 with both base editors, introducing a C64Y mutation with CBE, and C64R and L63P mutations with ABE (Supplementary Fig. 7h–k). This guide did not validate with NG-CBE, but did previously with WT-CBE[25] and NG-ABE. While we were unable to parse the effects of these individual mutants based on the spectrum of alleles in our data, C64R scored as LOF with SGE and L63P is pathogenic in ClinVar, so it is likely that both of these mutations contribute to the LOF phenotype.

All validation results are summarized in Fig. 6h. We identified 5 guides, which utilized intermediate or active PAMs, that introduced deleterious mutations with one or both base editors; 2 mutate the RING domain, and 3 the BRCT domain. 3 of 8 guides with intermediate or active PAMs introduced benign edits with both base editors, indicative of false positives in the primary screen. This aligns with our observations from previous WT-CBE BRCA1 and BRCA2 screen validations where 5/13 guides represented false positives from the primary screen[25]. None of the 10 guides that utilized an inactive PAM validated, and sequencing showed little editing at these sites, reinforcing the PAM-specificity of these Cas9 variants and highlighting the necessity of validating primary screening results to avoid drawing conclusions from off-target effects.

**Base editing of BCL2.** Since its FDA approval in 2016, Venetoclax, which targets the anti-apoptotic protein BCL2, has been administered to patients with chronic lymphocytic leukemia (CLL), small lymphocytic lymphoma (SLL), or acute myeloid leukemia (AML)[39]. Unfortunately, many develop resistance to treatment. Many of these tumors have single amino acid mutations in BCL2, and several other mutations that lead to drug resistance have been characterized in human cells or mice[40–42]. We set out to identify additional resistance-causing mutations in BCL2, as a better understanding of these mechanisms can improve patient monitoring, allow for tailored treatment plans, and inform the design of new, mutation-agnostic drugs.

We designed a tiling library targeting BCL2 with Cas9-NG, generated both CBE and ABE versions, and screened in triplicate at high coverage (>10,000 cells per sgRNA) in MOLM13, an AML line sensitive to BCL2 inhibition. We treated cells with 62.5 nM Venetoclax for 14 days (Fig. 7a). LFCs for untreated cells were calculated relative to pDNA, and LFCs for Venetoclax-treated cells were calculated relative to untreated arms (Supplementary Data 7). We calculated z-scores for each guide relative to intergenic controls. 16 guides enriched with a z-score >3 with either or both base editors (Supplementary Fig. 8a), and 68.8% (11/16) of these are predicted to edit between positions 100–175, a region containing the P2 and P4 pockets responsible for Venetoclax binding (Fig. 7b). Notably, several sites of resistance mutations observed clinically (G101, D103, F104) fall within this region[40–42].

**BCL2 Validation.** We chose five sgRNAs to validate with both base editors, including three guides predicted to make missense edits at residues 103-105 (sg19–21), and two guides predicted to make missense edits at residues 148/149 (sg22) and 169 (sg23), which are denoted on the 3D structure of BCL2 in complex with Venetoclax (Fig. 7c). We performed validation screens as described above, with each of the guides individually transduced into MOLM13 cells in duplicate. The conditions from the primary BCL2 tiling screens were replicated, and following isolation and amplification of genomic DNA, we deep-sequenced the targeted loci.

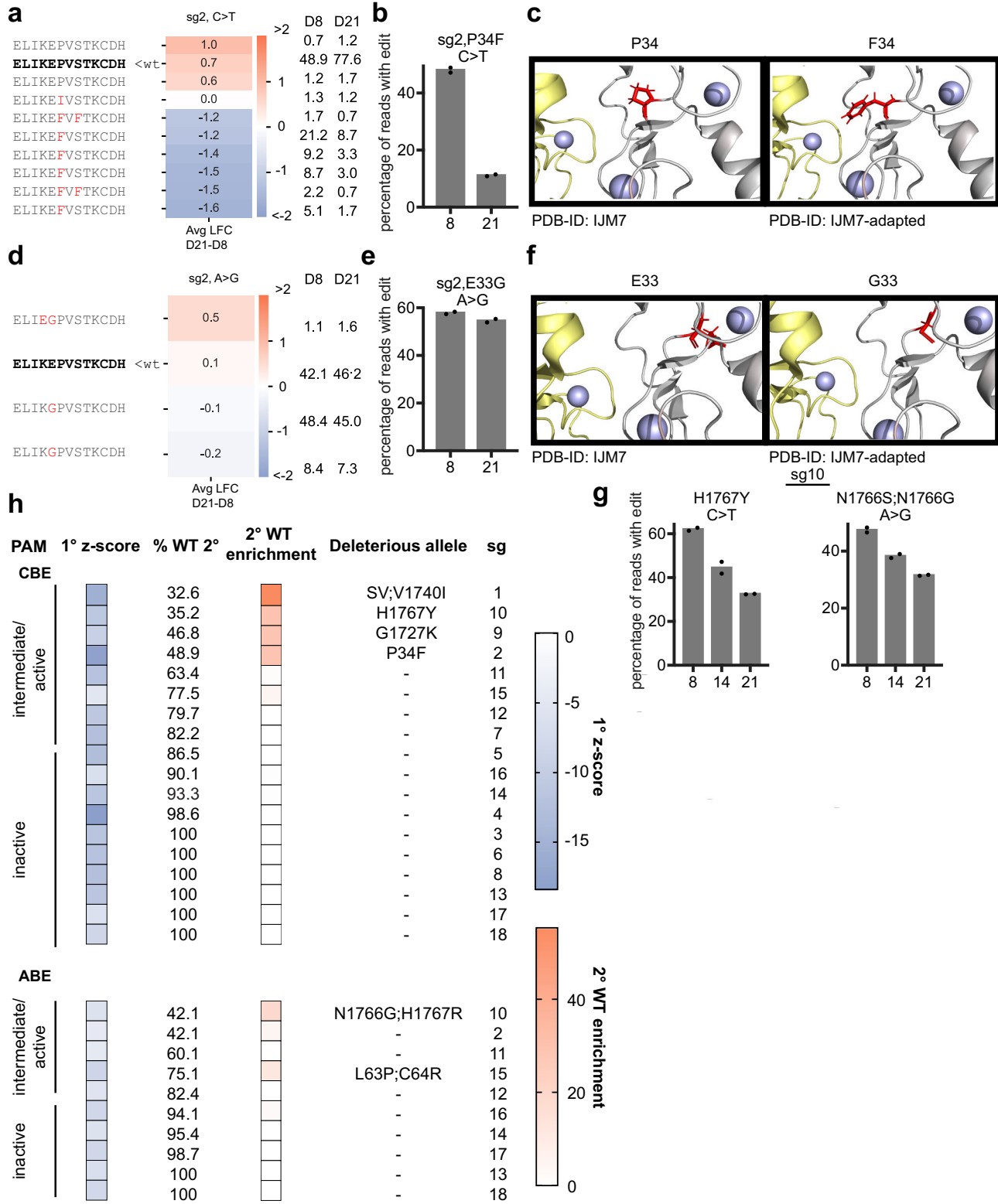

Editing levels at the early time point ranged from 11.7–55.4% with ABE and 10.9%–47.9% with CBE in the predicted window (Supplementary Fig. 8b). While all predicted edits were based on the canonical 4–8 window, we observed editing in the 3–10 window with both ABE and CBE, as well as low levels of C > T editing farther afield (sg22) which led to several unpredicted amino acid substitutions (Supplementary Fig. 8b). As before, we used the relative abundances of the WT allele at the early and late

time points to evaluate whether a guide validated. If the WT allele depleted by more than 10% under the selective pressure of Venetoclax, we considered it to be validated (Supplementary Fig. 8c). 8 of 8 guide-BE combinations validated as true positives, and 2 of 2 validated as true negatives. Further, sg20, 21, and 22 validated with both ABE and CBE, whereas sg19 and sg23 validated with the base editor with which they scored in the primary screens. With the non-scoring base editor, these sgRNAs

**Fig. 6 Validation of *BRCA1* hits identified by Cas9-NG. a** Translated sequence around the sgRNA for any allele with at least 1% abundance in any condition. The WT sequence is bolded in black, unchanged amino acids are in grey, and substitutions are highlighted in red. Avg LFC from day 21 to day 8 is indicated on the heatmap and relative percent abundance of each allele is indicated to the right (normalized after filtering for alleles with <1% abundance at both timepoints). **b** Percentage of all sequencing reads containing the indicated mutation at each timepoint. Dots indicate $n = 2$ biological replicates. **c** View of the RING domain (PDB IJM7) of *BRCA1* (grey) bound to the RING domain of BARD1 (yellow), with Zn2+ atoms in purple. The left panel shows the canonical amino acid residue in red, the right panel shows the structure with the P34F substitution. **d**, **e** Same as (**a**, **b**) for sg2 screened with ABE. **f** Same as (**c**), but with the E33G substitution. **g** Same as (**b**, **e**). **h** Summary of validation results. 1° z-score indicates the average z-scored LFC of the sgRNA in the primary screen. % WT 2° indicates the % of reads that were still WT (unedited) on day 8 of the validation experiment. 2° WT enrichment indicates the average change in the abundance of the WT allele from day 8 to day 21 in the validation experiment. PAM bin is indicated on the left. SV indicates "splice variant". Source data are provided as a Source Data file.

were predicted to make either a silent edit (sg23, CBE), or no edit (sg19, ABE). That top hits from the *BCL2* screens had a higher validation rate than those from the *BRCA1* screens is likely because the former is a positive selection screen, which presents fewer opportunities for off-target activity to score.

We next examined the enrichment of specific alleles to determine causal resistance mutations. With sg20 we saw strong enrichment for the D103E mutation caused by a C > A transversion at position 5 (Supplementary Fig. 8d). In this case, all missense mutations were the result of out-of-window edits or transversions, highlighting the necessity for direct sequencing of the edited locus. When paired with ABE, sg20 edited at A4, resulting in enrichment of the D103G allele, which likely disrupts the α2 helix (Fig. 7d–f). When we examined sg21 with CBE, we also saw enrichment for missense mutations at D103; substitution for an asparagine (D103N) was most favored upon Venetoclax treatment, but we also observed enrichment of the D103Y allele, as well as the dual replacement of D102 and D103 with asparagine (Supplementary Fig. 8e). The D103 residue falls within the P4 pocket and is important for hydrogen binding between the azaindole moiety of Venetoclax and BCL2[41]. Both D103E and D103Y have been previously recorded in patient samples bearing the G101V mutation[41,43], and Blombery and colleagues have shown that D103E mutagenesis causes the P4-binding pocket to more closely resemble that of BCL-xL, which is not inhibited by Venetoclax. With ABE, sg21 predominantly enriched for the F104L mutation that increases the P2-binding pocket volume[44] and likely disrupts a hydrogen bond between Venetoclax and the side chain of F104 (Fig. 7g, h, Supplementary Fig. 8f). For sg19 we saw C > T editing at positions 5, 7, 8, and 9, introducing an S105F missense mutation in 94.5% of edited alleles at the early time point (Supplementary Fig. 8g). In all cases this mutant enriched during treatment with Venetoclax, and dual editing of S105F and R106C enriched further still (Fig. 7i, j). Interestingly, the strongest LFC was seen with a rare in-frame deletion that removes R106.

The preceding three sgRNAs introduced edits at positions 102–106, which are located in the α2 helix. Edits at the α5 helix and the non-core α6 helix also enriched in the primary screen. With sg22 we observed several resistance mutations at positions 148–152. With CBE, an A149T mutation comprised 94.2% of all edited alleles on day 7 (Supplementary Fig. 8h), which alone was able to confer resistance, but when V148I occurred in combination, we saw further enrichment (Fig. 7k, l). When sg22 was screened with ABE, we saw A > G editing at positions 3, 4, 9 and 10, leading to a V148A substitution in all edited alleles. This edit alone was sufficient to cause Venetoclax resistance, and we observed secondary edits which also enriched during drug treatment (Supplementary Fig. 8i).

The final validated sgRNA (sg23) was predicted to make a silent edit with CBE, though we observed low levels of editing in a large window (C0-C18), resulting in non-resistant missense edits. With ABE all edited alleles carried the L169P missense mutation

(Supplementary Fig. 8j). Interestingly, this edit only enriched when V170A mutagenesis was observed in tandem (Fig. 7m). This resistance mechanism is particularly interesting, because residues 169 and 170 do not come in direct contact with Venetoclax. Mapping of these mutations onto the crystal structure of BCL2 shows the potential of a larger structural impact, whereby substitution with a significantly smaller side chain on the inner face of the helix, or disrupting the helix altogether with a proline substitution, may create vacated space that may cause additional conformational changes (Fig. 7n). A summary of the performance of sgs19−23 in both screens is provided (Supplementary Fig. 8k).

By leveraging PAM-flexible Cas9-NG paired with ABE and CBEs, we were able to densely tile *BCL2* and identify nucleotide substitutions that confer resistance to Venetoclax, including three previously-documented mutations (F104L, D103E, D103Y), and several resistant mutations that, to our knowledge, have not been reported. This screen demonstrates the power of tiling base editing screens in a positive selection setting, and identifies a condensed region of BCL2 (100−175) harboring many resistance mutations, which may be particularly interesting for follow up with more exhaustive forms of mutagenesis.

## Discussion

We established a pipeline that allows for the profiling of high-fidelity variants generated to mitigate off-target effects as well as PAM-flexible variants that increase the targeting range of Cas9. With respect to the former, we find that WT-Cas9 remains the best option for most screening applications. However, if off-target effects are of particular concern, then eCas9-1.1 is the best option of those tested here. Importantly, here we benchmark these variants in the context of genetic screens in which both Cas9 and the guide are delivered via lentivirus, and thus must complex together intracellularly. In cases where the protein is complexed with the guide in vitro, i.e. RNP delivery, the binding conditions are extraordinarily more favorable, which may aid in preserving on-target activity.

We also screened five PAM-flexible variants, three of which showed promising activity at non-canonical PAMs. We directly compared the off-target profiles of Cas9-NG and SpG, and found them nearly indistinguishable. Given that SpG shows higher activity at more PAM sites than Cas9-NG, with a comparable off-target profile, we recommend performing future base editing screens with SpG. Due to the relatively high prevalence of false-negatives with the technology, especially in negative-selection screens, the benefit of added depth is worthwhile, enabling the use of multiple unique guides to pinpoint regions of interest in a target locus. The recent development of a nearly-PAM-less Cas9 variant[18], as well as approaches to generate C > G edits[45–48], suggest that more editing outcomes and thus finer resolution will be possible for base editor screens. By highlighting specific protein regions of high value, densely tiled base editing screens can guide the creation of smaller, more focused ORF libraries, or SGE

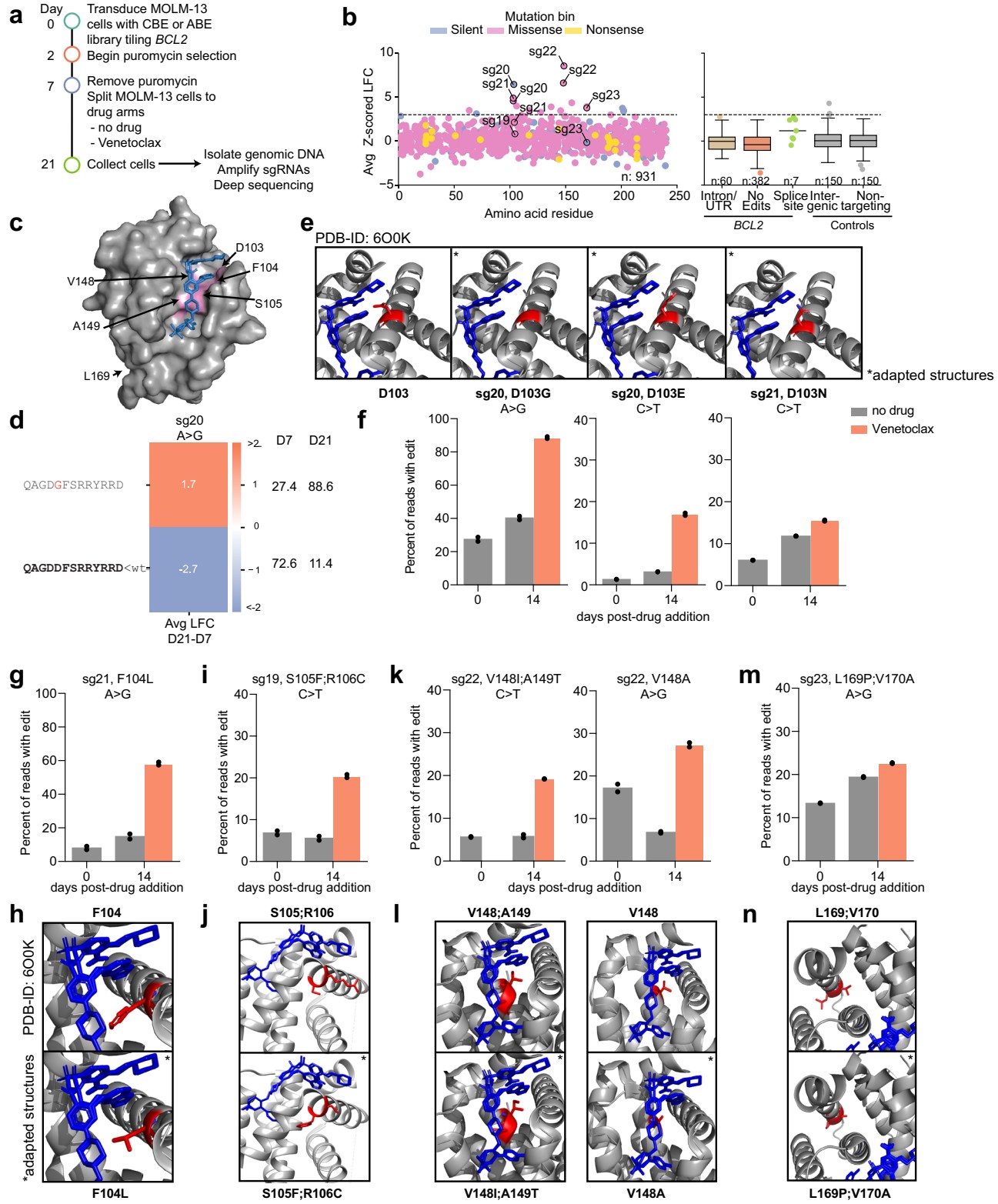

approaches, that are commensurately more efficient to screen[49–51]. The recent demonstration of prime editing technology for focused, saturating mutagenesis on haploidized loci[52] provides another potential path for follow-up of base editing screens.

Positive selection screens generally have lower false positive rates than negative selection screens[53], a trend consistent with the

validation rate of the *BCL2* and *BRCA1* screens presented here, as well as our previous drug resistance screens for inhibitors of *MCL1*, *BCL2L1*, and *PARP1*[25]. The low upfront costs of generating a pooled, base editing library, coupled with the relative ease of execution, suggest that such screens can be used to identify a resistance mutation that helps to prove the target of a less-characterized small molecule, as well as gain insight into

**Fig. 7 Venetoclax-resistant *BCL2* mutants identified by base editing with Cas9-NG. a** Timeline by which tiling screens were conducted. **b** Performance of sgRNAs targeting *BCL2* for the Venetoclax-treated arm, plotting both CBE and ABE screens, colored according to the predicted mutation bin. A dashed line delineates the z-score cutoff of 3. Boxes show the quartiles (Q1 and Q3) as minima and maxima and the center represents the mean; whiskers show 1.5 times the interquartile range (Q1-1.5*IQR and Q3 + 1.5*IQR). Categories with $n < 20$ are shown as individual dots. **c** 3D structure of BCL2 in complex with Venetoclax (PDB ID: 6O0K). Amino acids that sg19–23 are predicted to edit are highlighted in pink. **d** Translated sequence around sg20 for any allele with at least 1% abundance in any condition with ABE. The WT sequence is bolded in black, unchanged amino acids are in grey, and substitutions are highlighted in red. Avg LFC from day 21 to day 7 is indicated on the heatmap and relative percent abundance of each allele is indicated to the right (normalized after filtering for alleles with <1% abundance at both timepoints). **e** Structural visualization of WT D103 and the mutations indicated in (**f**). **f** Percentage of reads from the most enriched D103 mutants after 14 days of Venetoclax treatment. sgRNA, edit type, and amino acid mutation are indicated. Dots indicate $n = 2$ biological replicates. **g, i, k, m** Same as (**f**) but for indicated sgRNA, edit type, and position. **h, j, l, n** Same as (**e**) but showing the most enriched mutant indicated in **g, i, k**, or **m**, respectively. PDB-ID = 6O0K, adapted structures are indicated(*). Source data are provided as a Source Data file.

potential resistance mechanisms long before seeing what arises in patients. As cellular models are likely to already exist for probing the activity of such small molecules, there are few barriers to implementing such screens early in the drug discovery process.

## Methods

**Vectors.** pRosetta (Addgene 59700): lentiviral construct for expression of eGFP, puromycin resistance and blasticidin resistance.

pRosetta_v2 (Addgene 136477): modification of pRosetta to include a hygromycin resistance cassette; also known as pRDA_018.

pRDA_118 (Addgene 133459): U6 promoter expresses customizable SpCas9 guide; EF1a promoter provides puromycin resistance. This vector is a derivative of the lentiGuide vector, with a modification to the tracrRNA to eliminate a run of four thymidines.

pRDA_091: U6 promoter expresses customizable SpCas9 guide; EF1a promoter provides puromycin resistance. This vector also contains a Tet3G cassette that was not utilized in this study.

All Cas9 variants have: EF1a expresses Cas9; T2A site provides blasticidin resistance and P2A site provides mKate2. The Cas9 variants were generated by introducing the point mutations (Genscript) described in the original publications.

pRDA_085 (Addgene 158583): WT-Cas9.

pRDA_151 (Addgene 179084): Cas9-HF[8]. Point mutations: N497A/ R661A/ Q695A/ Q926A

pRDA_152 (Addgene 179085): eCas9-1.1[10]. Point mutations: K848A/ K1003A/ R1060A

pRDA_153 (Addgene 179086): evoCas9[12]. Point mutations: M495V/ Y515N/ K526E/ R661Q

pRDA_154 (Addgene 179087): xCas9-3.7[15]. Point mutations: A262T/ R324L/ S409I/ E480K/ E543D/ M694I/ E1219V

pRDA_155 (Addgene 179088): Cas9-VQR[13]. Point mutations: D1135V /R1335Q/ T1337R

pRDA_156 (Addgene 179089): Cas9-VRER[13]. Point mutations: D1135V /G1218R/ R1335E/ T1337R

pRDA_157 (Addgene 179090): HypaCas9[9]. Point mutations: N692A/ M694A/ Q695A/ H698A

pRDA_275 (Addgene 179091): Cas9-NG[14]. Point mutations: L1111R/ D1135V/ G1218R/ E1219F/ A1322R/ R1335V/ T1337R

pRDA_381 (Addgene 179092): HiFi Cas9[11]. Point mutations: R691A

pRDA_449 (Addgene 179093): SpG[18]. Point mutations: D1135L/ S1136W/ G1218K/ E1219Q/ R1335Q/ T1337R

All base editor constructs have: U6 promoter expresses customizable guide RNA with a 10x guide capture sequence at the 3′ end of the tracrRNA to facilitate future use with direct capture single cell RNA sequencing[54]; core EF1a (EFS) expresses codon-optimized ABE or CBE, and 2A site provides puromycin resistance. Note that the ABE8e constructs contain the V106W mutation.

pRDA_256 (Addgene 158581): WT-BE3.

pRDA_336 (Addgene 179095): NG-BE3.

pRDA_478 (Addgene 179096): SpG-BE3.

pRDA_426 (Addgene 179097): WT-ABE8e.

pRDA_429 (Addgene 179098): NG-ABE8e.

pRDA_479 (Addgene 179099): SpG-ABE8e.

**Cell lines and culture.** A375, MOLM13 and MELJUSO cells were obtained from Cancer Cell Line Encyclopedia at the Broad Institute. Unmodified HAP1 cells (item C631) were obtained from Horizon Discovery. HEK293Ts were obtained from ATCC (CRL-3216). MOLM13 cells were selected based on data from genome-wide CRISPR screens and cancer cell drug sensitivity screens (CTD^2 and GDSC) found on the Cancer Dependency Map Portal which identified MOLM13 as dependent on *BCL2*.

All cells regularly tested negative for mycoplasma contamination and were maintained in the absence of antibiotics except during screens, validation experiments, and lentivirus production, during which media was supplemented

with 1% penicillin-streptomycin. Cells were passaged every 2−4 days to maintain exponential growth and were kept in a humidity-controlled 37 °C incubator with 5.0% CO2. Media conditions and doses of polybrene, puromycin, blasticidin, and hygromycin were as follows, unless otherwise noted:

A375: RPMI + 10% fetal bovine serum (FBS); 1 μg/mL; 1 μg/mL; 5 μg/mL; N/A

HAP1: IMDM + 10% FBS; 4 μg/mL; 2 μg/mL; 5 μg/mL; N/A

HEK293T: DMEM + 10% heat-inactivated FBS; N/A; N/A; N/A; N/A

MELJUSO: RPMI + 10% FBS; 4 μg/mL; 1 μg/mL; 4 μg/mL; 100 μg/mL

MOLM13: RPMI + 10% FBS; 4 μg/mL; 1 μg/mL; N/A; N/A

**PAM-mapping library design.** 50 essential and nonessential genes were picked from prior screens performed in A375 and HT29. *BRCA1* and *BRCA2* were also included to increase the coverage per PAM sequence. sgRNA sequences tiling the coding sequence of the principal Ensembl transcript of these genes were designed. Four nucleotides following the sgRNA sequence were reported as the PAM sequence. The library was filtered to exclude any sgRNAs with BsmBI sites or a TTTT sequence. Promiscuous guides (defined as the PAM-proximal 18mer having > = 5 off-targets with up to 1 mismatch in the genome) were filtered out. We aimed to pick 50 sgRNAs per PAM sequence for the essential genes, *BRCA1*, and *BRCA2* and 25 sgRNAs per PAM sequence for nonessential genes. In doing so, we picked 10 G19, 30 g20 and 10 G20 sgRNAs for essential genes and 5 G19, 15 g20 and 5 G20 sgRNAs for nonessential genes.

**BRCA1 base editor tiling library design.** Guide sequences for tiling libraries were designed using sequence annotations from Ensembl (GRCg38). We used Ensembl's REST API (https://rest.ensembl.org/) to obtain the genomic locations of transcripts, transcript sequences, and protein sequences, and used these to annotate each sgRNA with its predicted edits. We included all sgRNAs targeting the coding sequence; we also included all sgRNAs for which the start was up to 30 nucleotides into the intron and UTRs. We designed every possible guide (using an NNNN PAM) against the longest annotated transcript for *BRCA1* (ENST00000471181, 1884 amino acids) using an editing window of 4−8 nucleotides for both CBE and ABE screens. We filtered out guides >5 perfect matches in the genome. The library was filtered to exclude any sgRNAs with BsmBI sites or a TTTT sequence. We used NCBI's ClinVar database (https://www.ncbi.nlm.nih.gov/clinvar/) to annotate the clinical significance of the variants introduced in this *BRCA1* base editing screens.

**BCL2 base editor tiling library design.** We used the Ensembl transcript ENST00000333681.5 to design all guides targeting *BCL2*, regardless of PAM, annotating edits based on an editing window of 4−8; we also included all sgRNAs for which the start was up to 29 nucleotides into the intron and UTRs. Guides with PAMs that scored as a fraction active ≤ 0.1 (from the PAM tiling screen) were filtered out for a total of $n = 96$ inactive PAMs in the library. Finally, the library was filtered to exclude any sgRNAs with BsmBI sites or a TTTT sequence.

**Library production.** Oligonucleotide pools were synthesized by CustomArray. BsmBI recognition sites were appended to each sgRNA sequence along with the appropriate overhang sequences (bold italic) for cloning into the sgRNA expression plasmids, as well as primer sites to allow differential amplification of subsets from the same synthesis pool. The final oligonucleotide sequence was: 5′-[Forward Primer]CGTCTCAC*ACCG*[sgRNA, 20 nt]*GTTT*CGAGACG[Reverse Primer]

Primers were used to amplify individual subpools using 25 μL 2x NEBnext PCR master mix (New England Biolabs), 2 μL of oligonucleotide pool (~40 ng), 5 μL of primer mix at a final concentration of 0.5 μM, and 18 μL water. PCR cycling conditions: (1) 98 °C for 30 s; (2) 53 °C for 30 s; (3) 72 °C for 30 s; (4) go to (1), x 24.

In cases where a library was divided into subsets, unique primers could be used for amplification:

Primer Set; Forward Primer, 5′ – 3′; Reverse Primer, 5′ – 3′

1; AGGGCACTTGCTCGTACGACG; ATGTGGGCCCGGCACCTTAA

2; GTGTAACCCGTAGGGCACCT; GTCGAGAGCAGTCCTTCGAC

3; CAGCGCCAATGGGCTTTCGA; AGCCGCTTAAGAGCCTGTCG

4; CTACAGGTACCGGTCCTGAG; GTACCTAGCGTGACGATCCG
5; CATGTTGCCCTGAGGCACAG; CCGTTAGGTCCCGAAAGGCT
6; GGTCGTCGGCATCACAATGCG; TCTCGAGCGCCAATGTGACG

The resulting amplicons were PCR-purified (Qiagen) and cloned into the library vector via Golden Gate cloning with Esp3I (Fisher Scientific) and T7 ligase (Epizyme); the library vector was pre-digested with BsmBI (New England Biolabs). The ligation product was isopropanol precipitated and electroporated into Stbl4 electrocompetent cells (Invitrogen) and grown at 30 °C for 16 h on agar with 100 µg/mL carbenicillin. Colonies were scraped and plasmid DNA (pDNA) was prepared (HiSpeed Plasmid Maxi, Qiagen). To confirm library representation and distribution, the pDNA was sequenced.

**Lentivirus production.** For small-scale virus production, the following procedure was used: 24 h before transfection, HEK293T cells were seeded in 6-well dishes at a density of $1.5 \times 10^6$ cells per well in 2 mL of DMEM + 10% heat-inactivated FBS. Transfection was performed using TransIT-LT1 (Mirus) transfection reagent according to the manufacturer's protocol. Briefly, one solution of Opti-MEM (Corning, 66.75 µL) and LT1 (8.25 µL) was combined with a DNA mixture of the packaging plasmid pCMV_VSVG (Addgene 8454, 250 ng), psPAX2 (Addgene 12260, 1250 ng)[55], and the transfer vector (e.g., pLentiGuide, 1250 ng). The solutions were incubated at room temperature for 20–30 min, during which time media was changed on the HEK293T cells. After this incubation, the transfection mixture was added dropwise to the surface of the HEK293T cells, and the plates were centrifuged at $1000 g$ for 30 min at room temperature. Following centrifugation, plates were transferred to a 37 °C incubator for 6–8 h, after which the media was removed and replaced with DMEM + 10% FBS media supplemented with 1% BSA. Virus was harvested 36 h after this media change.

A larger-scale procedure was used for pooled library production. 24 h before transfection, $18 \times 10^6$ HEK293T cells were seeded in a 175 cm$^2$ tissue culture flask and the transfection was performed the same as for small-scale production using 6 mL of Opti-MEM, 305 µL of LT1, and a DNA mixture of pCMV_VSVG (5 µg), psPAX2 (50 µg), and 40 µg of the transfer vector. Flasks were transferred to a 37 °C incubator for 6–8 h; after this, the media was aspirated and replaced with BSA-supplemented media. Virus was harvested 36 h after this media change.

**Determination of antibiotic dose.** In order to determine an appropriate antibiotic dose for each cell line, cells were transduced with the pRosetta or pRosetta_v2 lentivirus such that approximately 30% of cells were transduced and therefore EGFP + . At least 1 day post-transduction, cells were seeded into 6-well dishes at a range of antibiotic doses (e.g. from 0 µg/mL to 8 µg/mL of puromycin). The rate of antibiotic selection at each dose was then monitored by performing flow cytometry for EGFP + cells. For each cell line, the antibiotic dose was chosen to be the lowest dose that led to at least 95% EGFP + cells after antibiotic treatment for 7 days (for puromycin) or 14 days (for blasticidin and hygromycin).

**Small molecule doses in pooled screens.** For *BRCA1* screens in MELJUSO cells, cisplatin (BioVision, 1550) was diluted in 0.9% NaCl and was screened at 1 µM. For *BCL2* screens in MOLM13 cells, Venetoclax (Selleckchem, S8048) was diluted in DMSO and was screened at 62.5 nM.

**Determination of lentiviral titer.** To determine lentiviral titer for transductions, cell lines were transduced in 12-well plates with a range of virus volumes (e.g. 0, 150, 300, 500, and 800 µL virus) with 1 to $3 \times 10^6$ cells per well in the presence of polybrene. The plates were centrifuged at $640 \times g$ for 2 h and were then transferred to a 37 °C incubator for 4–6 h. Each well was then trypsinized, and an equal number of cells seeded into each of two wells of a 6-well dish. Two days post-transduction, puromycin was added to one well out of the two. After 5 days, both wells were counted for viability. A viral dose resulting in 30–50% transduction efficiency, corresponding to an MOI of ~ 0.35–0.70, was used for subsequent library screening.

**Derivation of stable cell lines.** In order to establish Cas9 variant expressing cell lines for screens with the PAM-mapping tiling library and both off-target libraries, A375 cells were transduced with either pRDA_085, pRDA_151-157, pRDA_275, pRDA_381 or pRDA_449 and successfully transduced cells were selected with blasticidin for a minimum of 2 weeks. Cells were taken off blasticidin at least one passage before transduction with libraries.

**Pooled screens.** For pooled screens, cells were transduced in 2–3 biological replicates with the lentiviral library. Transductions were performed at a low multiplicity of infection (MOI ~ 0.5), using enough cells to achieve a representation of at least 500 transduced cells per sgRNA assuming a 20-40% transduction efficiency. For the CRISPRko screens, cells were plated in polybrene-containing media with $3 \times 10^6$ cells per well in a 12-well plate. Because the titer of all-in-one base editor viruses was low, cells were plated in polybrene-containing media with $1.5 \times 10^6$ cells per well in a 12-well plate. Plates were centrifuged for 2 h at $640 \times g$, after which 2 mL of media was added to each well. Plates were then transferred to an incubator for 4-6 h, after which virus-containing media was removed and cells

were pooled into flasks. Puromycin was added 2 days post-transduction and maintained for 5–7 days to ensure complete removal of non-transduced cells. Upon puromycin removal, cells were split to any drug arms (each at a representation of at least 1000 cells per sgRNA) and passaged every 2-4 days for an additional 2 weeks to allow sgRNAs to enrich or deplete; cell counts were taken at each passage to monitor growth.

**Genomic DNA isolation and sequencing.** Genomic DNA (gDNA) was isolated using the KingFisher Flex Purification System with the Mag-Bind® Blood & Tissue DNA HDQ Kit (Omega Bio-Tek). The gDNA concentrations were quantitated by Qubit. For samples where genomic DNA was limiting, gDNA was purified prior to PCR using the Zymo OneStep PCR Inhibitor Removal Kit (Zymo), per the manufacturer's instructions.

For PCR amplification, gDNA was divided into 100 µL reactions such that each well had at most 10 µg of gDNA. Plasmid DNA (pDNA) was also included at a maximum of 100 pg per well. Per 96-well plate, a master mix consisted of 150 µL DNA Polymerase (Titanium Taq; Takara), 1 mL of 10x buffer, 800 µL of dNTPs (Takara), 50 µL of P5 stagger primer mix (stock at 100 µM concentration), 500 µL of DMSO (if used), and water to bring the final volume to 4 mL. Each well consisted of 50 µL gDNA and water, 40 µL PCR master mix, and 10 µL of a uniquely barcoded P7 primer (stock at 5 µM concentration). PCR cycling conditions were as follows: (1) 95 °C for 1 min; (2) 94 °C for 30 s; (3) 52.5 °C for 30 s; (4) 72 °C for 30 s; (5) go to (2), x 27; (6) 72 °C for 10 min. PCR primers were synthesized at Integrated DNA Technologies (IDT). PCR products were purified with Agencourt AMPure XP SPRI beads according to manufacturer's instructions (Beckman Coulter, A63880), using a 1:1 ratio of beads to PCR product. Samples were sequenced on a HiSeq2500 HighOutput (Illumina) with a 5% spike-in of PhiX.

**Validation experiments.** For validation experiments in which the target site was directly sequenced, individual sgRNAs were cloned into either pRDA_336 (NG-CBE) or pRDA_429 (NG-ABE) and made into lentivirus as described above. At least $1.5 \times 10^6$ cells were transduced in duplicate with a virus volume to obtain ~30–50% transduction efficiency and were selected with puromycin for 5−7 days to remove untransduced cells; puromycin doses were as described above. After puromycin selection was removed, cells were split into any drug arms and cultured for an additional 14 days. Cell pellets were collected on days 7, 14, and 21 (*BCL2*) or 8, 14, and 21 (*BRCA1*).

Genomic DNA was isolated using either the Kingfisher as described above, or cells were lysed in 96-well plates using 25 µL per well of Lucigen QuickExtract DNA Extraction Solution (QE0905T). Briefly, 25 uL of lysis buffer was added to each well, plate was sealed and vortexed, then heated at 65 °C for 15 min, heated at 95 °C for 5 min and then stored at −20 °C. Target sites were amplified using a 2-step PCR. For the samples in which gDNA was isolated using the Kingfisher, in the first round of PCR, genomic DNA was amplified using custom primers designed to amplify each target site (see Supplementary Data 8). Each well contained 50 µL of NEBNext High Fidelity 2X PCR Master Mix (New England Biolabs), 0.5 µL of each primer at 100 µM, and 49 µL of gDNA. We used a touchdown PCR with the following cycling conditions: (1) 98 °C for 1 min; (2) 98 °C for 30 s; (3) 68 °C for 30 s (−1° per cycle); (4) 72 °C for 1 min; (5) Go to step 2, x 15; (6) 72 °C for 10 min. For samples subjected to the 96-well plate lysis, in the first step, the master mix for each 96-well plate consisted of: 75 µL Titanium Taq polymerase, 500 µL 10X Titanium Taq buffer, 400 µL dNTPs, 250 µL DMSO, 25 µL forward primer at 100 µM, 25 µL reverse primer at 100 µM, and water to bring the final volume to 10 mL. Forward and reverse primers were as described in Supplementary Data 8. Each well consisted of 5 µL crude lysate, 25.5 µL master mix, and water to 50 µL final volume. PCR cycling conditions were as follows: (1) 95 °C for 5:00, (2) 94 °C for 0:30, (3) 53 °C for 0:30, (4) 72 °C for 0:20, (5) Go to step 2, x 17 cycles, (6) 72 °C for 10:00.

The second round of PCR was the same for both approaches. It appended Illumina adapters and well barcodes for sequencing using the P5 primer "Argon" and the P7 primer "Kermit". Each well contained 1.5 µL of Titanium Taq (Takara), 10 µL of Titanium Taq buffer, 8 µL of dNTPs, 5 µL of DMSO, 0.5 µL of P5 primer at 100 µM, 10 µL of P7 primer at 100 µM, 55 µL of water, and 10 µL of PCR product from the first PCR. The following cycling conditions were used: (1) 95 °C for 1 min; (2) 94 °C for 30 s; (3) 52.5 °C for 30 s; (4) 72 °C for 30 s; (5) go to (2), x 15; (6) 72 °C for 10 min. Samples were pooled and purified by primer pair with Agencourt AMPure XP SPRI beads according to the manufacturer's instructions (Beckman Coulter, A63880), using a 1:1 ratio of beads to PCR product. DNA concentration was quantified using a Qubit and purified samples were pooled proportionally to their concentrations. The pooled library was quantified by Qubit and sequenced using the Illumina MiSeq with a 300 nucleotide single read and a 10% PhiX spike-in.

**Assessment of Cas9 expression levels.** Fresh virus for WT-Cas9 and each of the ten SpCas9 variants was prepared in parallel for this analysis. A375 cells were transduced with virus for each of the 11 constructs separately; 2 d after transduction, cells were selected with blasticidin (5 µg/mL) for 14 d. Cells were visualized by flow cytometry on the CytoFLEX S Sampler. To prepare samples for visualization, cells were fixed and permeabilized using the Abcam Cell Fixation and

Permeabilization Kit (Abcam, catalog no. ab185917). Cells were stained as per the kit's protocol using Cas9 Mouse mAB Alexa Fluor 647 Conjugate (Cell Signaling Technology, catalog no. 48796 s), diluted 1:100.

Cells were washed with PBS two additional times to remove residual antibody and were resuspended in flow buffer (PBS, 2% FBS, 5 μM EDTA). Alexa Fluor 647 signal was measured in the APC-A channel. mKate2 signal was measured in the PE-A channel. Flow cytometry data were analyzed using FlowJo (v10.8,1). Gates were set such that ~1% of cells score as APC-positive or PE-positive in the control condition (stained A375 parental cells).

### Quantification And Statistical Analysis

*Screen analysis.* Guide sequences were extracted from sequencing reads by running the PoolQ tool with the search prefix "CACCG" (https://portals.broadinstitute.org/gpp/public/software/poolq). Reads were counted by alignment to a reference file of all possible guide RNAs present in the library. The read was then assigned to a condition (e.g. a well on the PCR plate) on the basis of the 8 nt index included in the P7 primer. Following deconvolution, the resulting matrix of read counts was first normalized to reads per million within each condition by the following formula: read per guide RNA / total reads per condition x 1e6. Reads per million was then log2-transformed by first adding one to all values, which is necessary in order to take the log of guides with zero reads.

Prior to further analysis, we filtered out sgRNAs for which the log-normalized reads per million of the pDNA was > 3 standard deviations from the mean. We also filtered out any sgRNAs containing more than 5 off-target sites in the human genome (for non-NGGN guides) or containing more than 5 off-target sites in the human genome with a CFD score of 1.0 (indicating a perfect or near-perfect match) for NGGN guides. We then calculated the LFC between conditions. All dropout (no drug) conditions were compared to the plasmid DNA (pDNA); drug-treated conditions were compared to the time-matched dropout sample; with the exception of MELJUSO cells in the *BRCA1* screens, which were compared to the pDNA because loss of *BRCA1* had some viability effect in the absence of drug. We assessed the correlation between LFC values of replicates.

*Analysis of PAM-mapping screens.* The initial PAM characterization screens were carried out in four rounds, each sequenced separately. pDNA raw reads were summed across screens, and log-normalized. Because *BRCA1* and *BRCA2* are not widely panlethal, guides targeting these genes were excluded from all analyses downstream of the calculation of LFC. Precision-recall was calculated using sgRNAs targeting essential genes as positive controls, and nonessential genes as negative controls. Fraction active was calculated by quantifying the fraction of guides targeting essential genes that were more depleted than the 5th percentile of the most active nonessential guides with the same PAM, and all non-targeting guides. ROC-AUC was calculated using guides targeting essential genes as positive controls, and guides targeting nonessential genes as negative controls.

*Off-target analyses.* Using the perfect match guides as true positives and the 1000 control guides as true negatives, we fit a logistic regression for each condition to predict whether each guide is a perfect match or control based on its LFC. We then used the fit model to map from log2-fold-changes to the probability of being active (i.e. being a perfect match sgRNA). A value near 1 indicates a guide is active and a value near 0 indicates a guide is inactive. These values were then used to calculate the CFD scores for all enzymes screened (provided in Supplementary Data 4). For example, if the interaction between the sgRNA and DNA has a single rG:dA mismatch in position 6, then that interaction receives a score of 0.67. If there are two or more mismatches, then individual mismatch values are multiplied together. For example, an rG:dA mismatch at position 7 coupled with an rC:dT mismatch at position 10 receives a CFD score of $0.57 \times 0.87 = 0.50$. For the high-fidelity off-target screens, these data were also used to predict the activity at double mismatches. Using the same logistic regression model as described above, we defined any double mismatch guide with a > 50% probability of being as active as a perfect match based on its LFC as active. We used this cutoff to define true positives and ranked the multiplied single mismatch probabilities to predict the activity of the double mismatch guides. For the off-target screens with Cas9-NG and SpG, data were first filtered for PAMs that were intermediate/active with either enzyme and the top 18 perfect match guides (even if they didn't have an intermediate/active PAM) were maintained.

*Base editing analyses.* To obtain a "mutation bin" for each sgRNA, we ordered the mutation types as: Nonsense > Splice site > Missense > Intron > Silent > UTR. Guides containing multiple mutation types were binned as the most severe mutation type. Guides predicted to make no edits in the editing window were binned as "No edits". To obtain a "clinical significance bin," (for *BRCA1* only) we classified sgRNAs predicted to introduce multiple ClinVar SNPs based on the most severe clinical significance: Pathogenic > Likely pathogenic; Pathogenic / Likely pathogenic > Uncertain significance > Conflicting reports of pathogenicity > Variant not listed in ClinVar > Likely benign; Benign / Likely benign > Benign. With this ordering, sgRNAs were only binned as "Likely benign" or "Benign" if they did not introduce any mutations not listed in ClinVar, which effectively have an unknown functional significance.

For the calculations regarding the number of targetable residues in *BRCA1* shown in Fig. 5a, we considered guides included in the libraries we screened with (filtered for off-targets, BsmBI sites and 4Ts targeting *BRCA1 n* = 11,524) with PAMs that were intermediate/active with variant base editors or active (NGGN) for WT base editors. For the combination of ABE and CBE, we took the unique set of the sum of residues editable with either CBE or ABE.

*Analysis of deep sequencing data from validation experiments.* CRISPResso2 (version 2.0.30) was used to process all sequencing reads from validation experiments[56]. CRISPResso2 was run in base editor mode using the default settings with the following changes: --min_average_read_quality 25, --w 20. We also set custom values for each sgRNA for --wc, --exclude_bp_from_left, --exclude_bp_from_right, and --default_min_aln_score; these parameters can be found in Supplementary Data 8.

To calculate replicate correlations, we used the "Alleles_frequency_table_around_sgRNA" file from the CRISPResso2 output, which contains the read counts for each allele (defined as a subsequence around the sgRNA). We then log-normalized the read counts for each sample (using the same formula described in the "Screen analysis" section). Finally, we filtered out any alleles with < 100 reads in all replicates and drug conditions for that sgRNA, and calculated the Pearson correlation between log-normalized reads.

For further analysis of alleles, we added an additional filter based on percentage frequency in all replicates and drug conditions for that sgRNA in order to avoid spurious LFC values due to low relative abundance: we filtered out any alleles that comprised < 1% of the total reads in *BRCA1*, and alleles that comprised < 2% of the total reads in *BCL2* (due to lower quality of the *BCL2* sequencing run).

*External datasets.* The data used for the comparison to our previously generated WT-Cas9 CFD shown in Supplementary Fig. 2b were obtained from the original publication[6]. For the comparison to Legut et al., we obtained data from the original publication[16] and considered the CD45 and CD55 data, excluding guides targeting the promoter region. We calculated LFCs by subtracting the bottom_bin from the top_bin for each Cas, z-scored these data based on the non-targeting controls, then calculated the median z-score of each 3 nucleotide PAM for both this, and our dataset. The data used for the comparison to Kim et al. were obtained from the original publication[23]. We used the average indel frequencies at target sequences grouped by 4-nucleotide PAM in the comparisons shown in Supplementary Fig. 3a–c. Saturation genome editing data was accessed from the original publication[35]. Because the libraries described in our study and the SGE study were designed against different transcripts (SGE used ENST00000357654), we converted the amino acid positions in the SGE data to be consistent with the transcript we designed against for the comparisons between our validation data and this gold standard dataset.

*Data visualization.* Figures were created with Python3 and GraphPad Prism (version 8). Schematics were created with BioRender.com. PyMOL (version 2.3.2) was used to map the screening data onto the following crystal structures from the Protein Data Bank: IJM7 (*BRCA1/BARD1* RING-domain heterodimer) and 6O0K (*BCL2* bound to Venetoclax). The following commands were used to visualize the *BRCA1/BARD1* RING-domain heterodimer in PyMOL (from Walton et al. 2020): cmd.set("bg_rgb", 'white') cmd.set("ambient", '0.21000') cmd.set("direct", '0.40000') cmd.set("reflect", '0.43000') cmd.set("power", '2.00000') cmd.set("spec_reflect", '-0.01000') cmd.set("line_width", '3.00000') cmd.set("cache_display", 'off') cmd.set("shininess", '30.00000') cmd.set("cartoon_sampling", '7') cmd.set("cartoon_loop_radius", '0.15000') cmd.set("cartoon_oval_length", '1.00000') cmd.set("auto_color_next", '1') cmd.set("max_threads", '4') cmd.set("specular_intensity", '0.30000') cmd.set("button_mode_name", '3-Button Viewing') cmd.set("seq_view", 'on') cmd.set("cartoon_ring_mode", '3')

**Statistical analysis**. All z-scores and Pearson correlation coefficients were calculated in Python.

**Reporting summary**. Further information on research design is available in the Nature Research Reporting Summary linked to this article.

## Data availability

Source data are provided with this paper. The read counts for all screening data and subsequent analyses are provided as Supplementary Data. Fastq files are deposited with the Gene Expression Omnibus (GSE180351) and the Sequence Read Archive (PRJNA753064). Source data are provided with this paper.

## Code availability

All custom code used for analysis and example notebooks are available on GitHub: https://github.com/gpp-rnd/cas9-variants-manuscript.

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

## Acknowledgements

We thank Amy Goodale, Briana Fritchman, Edith Sawyer, Hinako Kiwabe, Luke Sprenkle, Pema Tenzing, Tashi Lokyitsang and Xiaoping Yang for producing guide libraries and lentivirus; Olivia Bare, Yenarae Lee, and Max Macaluso for logistics support; Matthew Greene, Bronte Wen, Adam Brown, Doug Alan, Mark Tomko, and Tom Green for software engineering support; the Broad Institute Genomics Platform Walk-up Sequencing group for Illumina sequencing. We thank Marissa Feeley, Sarah Weiss, Nicky Persky, Dave Root, Kathleen Kristie, Russell Walton, Sumaiya Iqbal, and Benjamin Kleinstiver for helpful discussions.

## Author contributions

Conceived of the study: AKS, JGD. Executed genetic screens: AKS, ALG, ZMS, AVM, REH. Performed analyses: AKS, ALG, PR, ZMS, PCD, MH. Created visualizations: AKS, ALG, ZMS, PCD, AVM. Designed libraries: AKS, MH, ALG. Curated data: AKS, ALG, ZMS, MH. Wrote the manuscript: AKS, ALG, JGD. Supervised the project: JGD.

## Competing interests

JGD consults for Microsoft Research, Agios, Phenomic AI, Maze Therapeutics, BioN-Tech, and Pfizer; JGD consults for and has equity in Tango Therapeutics. JGD's research is supported in part by the Functional Genomics Consortium (Merck, Abbvie, Janssen, Vir, and Bristol Meyers Squibb). JGD's interests were reviewed and are managed by the Broad Institute in accordance with its conflict of interest policies. All other authors declare no competing interests.
