## [Peer Review File · Nature Communications]

Reviewers' Comments:

Reviewer #1:

Remarks to the Author:

In the study by Sangree et al., the authors present a PAM-mapping strategy that they used to characterise PAM-dependencies of various SpCas9 variants. By means of precision-recall analyses, focussed gRNA libraries and screens, as well as computational analyses (computing the probably for being active at off-target sites (including single and double mismatches)), the authors characterised on- and off-target activities of selected high-fidelity and PAM-flexible SpCas9 variants. With these experiments/analyses, the authors demonstrated that the two PAM-flexible SpCas9 variants NG and SpG have highly-flexible PAM sequences, while having good or excellent separation between perfectly matched and single or double mismatched gRNAs, respectively. Due to the increased target space of NG and SpG variants, the authors moved on to compare WT-SpCas9 with these variants in adenine/cytosine-base editor (A/CBE) applications, using the two recently described A/CBEs BE3.9max and ABE8e. In more detail, the authors applied the collection of SpCas9 variant-A/CBEs to the BRCA1 gene, for which the authors previously did a CBE screen with WT-SpCas9, as well as the clinically relevant gene-drug combination of BCL2 and Venetoclax. As for BRCA1, the authors identified depleted gRNAs in the functionally relevant RING and BRCT domains, a finding that is also consistent with the literature as well as their previous findings. As for BCL2, the authors identified the region containing the two Venetoclax-binding pockets P2 and P4 to be enriched for gRNAs conferring resistance to Venetoclax, and validation experiments with enriched-gRNAs from the primary screen revealed previously known resistance-causing mutations, as well as hitherto undescribed Venetoclax-resistance mutations.

While the study appears interesting at first, a closer look leaves the impression of an assembly of experiments that are forcefully connected by technological reasoning rather than biological questioning. Large parts of the work and results, e.g., PAM preferences, off-target activity, as well as the A/CBE BRCA1 study are already common knowledge and thus do not contribute to the presented novelty. A fact that is also acknowledged by the authors at multiple locations within the manuscript (lines 140, 142, 172, 201, 225, 236, 247, 343, 357, 379), leaving this reviewer with the impression that the presented work contains little, if any, noteworthy new results. Moreover, it appears that the logical flow and individual paragraphs are disconnected. For example, on-target activity was mapped for WT-Cas9, eSpCas9-1.1, Cas9-HF1, HypaCas9, and evoCas9; however, off-target activity was only mapped for WT-Cas9, eCas9-1.1, and HiFi-Cas9. This leaves the impression of inconsistency, as the authors do not provide any logical or result-based justification for excluding/including various Cas9 variants in different experimental conditions. For example, the authors do not provide on-target data for HiFi-Cas9, but the enzyme suddenly "pops up" at the off-target analysis without reasoning. Also, starting at line 308, the authors elaborate on the extended target range of PAM-flexible Cas9 variants for base-editing applications, but then fall short on integrating this information as well as their above-described PAM mapping information for their base-editor library designs, rendering the entire paragraph between line 308 and 315 useless, and overall questioning the usability of the presented approach (i.e., why was it even performed if it is not used later?). In line with this, NG and SpG Cas9 variants and their PAM flexibilities have previously been reported, and the presented PAM-mapping criteria for both enzymes have not been implemented in the presented experimental design of base-editor screens (all PAMs in BRCA1 and BCL2 were targeted); thus, the manuscript's title is misleading and factually incorrect. All these factors negatively interfere with reading and understanding the manuscript.

Despite these major issues that prevent this reviewer from positively moving forward with this manuscript, several additional concerns exist:

- 1) The wide range of correlations (0.25 to 0.87) for Cas9 variants during PAM mapping is highly concerning. Regardless of the activity of an enzyme, this reviewer would expect that replicates should overall correlate well. As the authors do not provide reasoning for the nature of this low reproducibility, this fact is concerning and questions downstream analyses and conclusions, at least from the low-reproducing screens.
- 2) The authors computed precision recalls for essential/non-essential genes to interpret the PAM-mapping results of Cas9 variants. While this analysis is well established for comparing screen performance; at this location, however, this is an unfitting metric for comparing different Cas9 variants/screens, as the essential/non-essential genes were originally determined by WT-SpCas9 and not the other variants. In other words, different Cas9 variants may reveal a different

set of essential/non-essential genes due to different biological requirements and characteristics. A fact that is also supported by the recent literature in which screening time as a sole changeable parameter contributed to vastly different sets of essential genes (Rahman et al., 2021).

3) The authors claim that WT-Cas9 does not show a preference for G19. However, when looking carefully into the associated figure, WT-Cas9 also shows a preference for G19 (although marginal), an effect that is likely masked by WT-Cas9's overall high performance/precision recall.

4) To assess the off-target activity of Cas9 variants, the authors did functional screenings in A375 cells. However, it is unclear if the authors used a reference genome sequence or the genomic sequence of the A375 cells to annotate and design perfect-matching gRNAs or gRNAs with single or double mismatched. Since mismatch tolerance is a very important characteristic of Cas9 variants that may even contribute to the PAM-mapping data, the author should clarify/discuss which strategy was used and reason why.

5) To assess the off-target activity of PAM-flexible Cas9 variants, the authors designed and screened a gRNA library containing perfect matched gRNAs, in addition to gRNAs with single and double mismatches. However, it is entirely unclear to this reviewer why only 50% of the perfectly-matching gRNAs were considered for downstream analyses. If only 50% performed well (argument of the authors for focusing on the 147 gRNAs), it raises the concern about the accuracy of the previously reported PAM-preferences for these variants. In line with this, the downstream result of good to perfect separation from single and double mismatches is thus not surprising, as the input data are highly biased towards performing gRNAs.

6) In the BRCA1 screen, the authors examined base editing activity by the separation of negative and positive controls, but fail to provide a quantitative value for screen/editing performance. This, however, has recently been done for knockout screens by computing the effect size (Cohen's *d*), a measure the authors should also consider to use.

7) The high false positive rates (3 out of 8 (37.5%), 2 out of 10 (20%)) in their BRCA1 and BCL2 validation experiments is concerning, and the authors should comment on this finding and discuss how this compares to WT-Cas9.

8) In the BCL2 validation, the authors reason that D103 mutations disrupts the alpha2 helix in the P4 pocket, a known and essential residue in Venetoclax binding. While this conclusion is intuitive, this entire part does not add to the presented result and should be moved to the conclusion/discussion section. In fact, each individual structural reasoning part (also for the other mutations) lacks experimental proof and should therefore be moved to the discussion. Moreover, the authors should clearly mark the presented structures with their associated PDB-ID references as well as "adapted structures (not experimentally confirmed)".

Minor issues:

- 1) Inconsistent use of WTCas9 and WT-Cas9 in text and figures.
- 2) Fig 1e/f - why are only 5 variants shown?
- 3) Fig 1d, 2b, 4b, SF4b, SF5d, - Y axis of density plots is missing.
- 4) Sup Fig 2 - 1) Typo: "work6", 2) y-axis of kde plot missing.
- 5) Line 205 - consistency of language: "showed intermediate activity" was "diminished" activity before.
- 6) Fig 3a - This appears to be a redundant graphic, i.e. what is the difference to Sup Fig 1a?
- 7) Sup Fig 3 text - "Kim at el31" reference is misspelled.

Reviewer #2:

Remarks to the Author:

The manuscript "Benchmarking of SpCas9 variants enables deeper base editor screens of BRCA1 and BCL2" by Annabel Sangree and colleagues describes the assessment of on- and off-target activities of SpCas9 variants based on their performance in pooled knock-out screens. From this benchmarking, the authors identify two variants with relaxed PAM requirements that are then used to demonstrate deeper tiling of endogenous genes in base editor screens.

General remarks.

An independent, side-by-side evaluation of SpCas9 variants as presented here is of interest to

both the screening- and (to an extent) the precision editing communities. Base editor screens have shown great potential for a number of important applications. Their performance is proportional to their targeting resolution. The characterization of PAM-relaxed Cas9 versions in this context is thus important. Overall, the paper is comprehensive, the experiments and the results are well presented and the methods are well described. The conclusions are supported by the data and relevant literature is cited. The work is in my opinion suitable for publication in Nature Communications, both in terms of quality and impact.

Major points.

1) My only major concern is the comparability of the cell lines containing the SpCas9 variants. Different virus batches can have different titers, potentially resulting in different average copy-number of SpCas9 in the different lines, which would affect Cas9 protein levels. In addition, protein stability of the variants might differ. A fair benchmarking would require comparable expression levels of all variants. Thus, protein expression of the different SpCas9 variants in the screened cell lines has to be assessed. If significant differences are found, the results need to be re-interpreted accordingly.

Minor points.

- 1) The abstract states that 11 variants were assessed, Figure 1 has only 9 variants + WTCas9.
- 2) Throughout, it is not easy to follow which variants were assessed with which assays. A table summarizing this information would help. Please also explain why variants were included/excluded.

Point-by-point rebuttal

Please note that during code review, we implemented a small change in the analysis of our base editor validation data related to the method for processing and filtering sequencing reads, which impacted three of the BRCA1 samples. We have updated the corresponding figures (Fig 6h and Supplementary Fig 7b&c). This resulted in minor changes in the %WT allele values for sg14 (CBE), sg17 (CBE) and sg17 (ABE). These have no effect on the interpretations presented in this manuscript.

Reviewer #1

In the study by Sangree et al., the authors present a PAM-mapping strategy that they used to characterise PAM-dependencies of various SpCas9 variants. By means of precision-recall analyses, focussed gRNA libraries and screens, as well as computational analyses (computing the probability for being active at off-target sites (including single and double mismatches)), the authors characterised on- and off-target activities of selected high-fidelity and PAM-flexible SpCas9 variants. With these experiments/analyses, the authors demonstrated that the two PAM-flexible SpCas9 variants NG and SpG have highly-flexible PAM sequences, while having good or excellent separation between perfectly matched and single or double mismatched gRNAs, respectively. Due to the increased target space of NG and SpG variants, the authors moved on to compare WT-SpCas9 with these variants in adenine/cytosine-base editor (A/CBE) applications, using the two recently described A/CBEs BE3.9max and ABE8e. In more detail, the authors applied the collection of SpCas9 variant-A/CBEs to the BRCA1 gene, for which the authors previously did a CBE screen with WT-SpCas9, as well as the clinically relevant gene-drug combination of BCL2 and Venetoclax. As for BRCA1, the authors identified depleted gRNAs in the functionally relevant RING and BRCT domains, a finding that is also consistent with the literature as well as their previous findings. As for BCL2, the authors identified the region containing the two Venetoclax-binding pockets P2 and P4 to be enriched for gRNAs conferring resistance to Venetoclax, and validation experiments with enriched-gRNAs from the primary screen revealed previously known resistance-causing mutations, as well as hitherto undescribed Venetoclax-resistance mutations.

While the study appears interesting at first, a closer look leaves the impression of an assembly of experiments that are forcefully connected by technological reasoning rather than biological questioning.

Large parts of the work and results, e.g., PAM preferences, off-target activity, as well as the A/CBE BRCA1 study are already common knowledge and thus do not contribute to

the presented novelty. A fact that is also acknowledged by the authors at multiple locations within the manuscript (lines 140, 142, 172, 201, 225, 236, 247, 343, 357, 379), leaving this reviewer with the impression that the presented work contains little, if any, noteworthy new results.

We believe that our manuscript contains several noteworthy new results. To our knowledge, no one has performed an ABE-based screen targeting BRCA1, so these results cannot be, to use the reviewer's phrase, "common knowledge." Nor are there any reports of base editor screens using expanded PAM Cas9 variants to target BRCA1 or any other gene. Nor have the off-target profiles of SpG and NG been assessed to this depth, with thousands of guides, and certainly not head-to-head. Nor are there any screens using base editors of any kind targeting BCL2, and we identify several new venetoclax-resistant mutations in these screens.

As for our characterization complimenting efforts from others, we note that we explored how these enzymes perform when integrated as single copy lentiviruses and targets are endogenous genes, while most other studies have opted for transient overexpression via transfection and the use of reporter constructs rather than endogenous gene targets. Thus this work helps to establish the performance of these enzymes under screening conditions.

Finally, we do not believe that manuscripts driven by technological reasoning are inherently less valuable additions to the literature than those driven by biological questioning. The purpose of this manuscript is to benchmark technology and describe the benefits and limitations of a particular experimental approach, in this case, understanding variants through base editor screens. The manuscript provides an expectation of how such screens are likely to perform in one's own hands, guidance on what are useful directions to pursue, and probably just as importantly, what not to pursue.

Moreover, it appears that the logical flow and individual paragraphs are disconnected. For example, on-target activity was mapped for WT-Cas9, eSpCas9-1.1, Cas9-HF1, HypaCas9, and evoCas9; however, off-target activity was only mapped for WT-Cas9, eCas9-1.1, and HiFi-Cas9. This leaves the impression of inconsistency, as the authors do not provide any logical or result-based justification for excluding/including various Cas9 variants in different experimental conditions.

For example, the authors do not provide on-target data for HiFi-Cas9, but the enzyme suddenly "pops up" at the off-target analysis without reasoning.

In line 156 of the original version of the manuscript, we wrote:

We performed screens in duplicate in A375 cells stably expressing three different variants: WT-Cas9; eCas9-1.1, as this was the best-performing variant in the PAM-mapping library, and HiFi Cas9, another recently described variant we had not previously assessed.

We have now clarified (in the legend for Figure 1a) that our initial clone of HiFi-Cas9 contained a mutation, and thus it was not included in the first experiments. Additionally, in response to a comment from reviewer 2, we have provided an organizational figure (Fig 1a) to help readers navigate what screens were done with which Cas variants and libraries.

Also, starting at line 308, the authors elaborate on the extended target range of PAM-flexible Cas9 variants for base-editing applications, but then fall short on integrating this information as well as their above-described PAM mapping information for their base-editor library designs, rendering the entire paragraph between line 308 and 315 useless, and overall questioning the usability of the presented approach (i.e., why was it even performed if it is not used later?). In line with this, NG and SpG Cas9 variants and their PAM flexibilities have previously been reported, and the presented PAM-mapping criteria for both enzymes have not been implemented in the presented experimental design of base-editor screens (all PAMs in BRCA1 and BCL2 were targeted); thus, the manuscript's title is misleading and factually incorrect. All these factors negatively interfere with reading and understanding the manuscript.

It has been our experience that there is not a meaningful difference in the difficulty or cost of screening a library in the 1,000 - 10,000 guide range, in which case there is no meaningful downside to including more guides, such as guides that are unlikely to be effective because they utilize an inactive PAM. Further, this also future-proofs the libraries for additional variants that recognize other PAM sequences, as we now note in the text.

Rather, we believe that their inclusion further strengthens our conclusions about best-practices going forward. Indeed, if even a small number of guides with non-NG PAMs scored and validated, especially in the BCL2/venetoclax screen which, as a positive selection screen, is more sensitive to rare events, that would be useful information for readers to know about when thinking about what guides to include or exclude in tiling libraries targeting their genes of interest. Including all guides thus *strengthens* the conclusions of the manuscript of what PAMs are useful to include in such libraries. We have added a sentence to the manuscript clarifying this logic.

As to the title of the manuscript, we did in fact benchmark SpCas9 variants, and then acquired more information about BRCA1 and BCL2 by using base editors with these variants than we would have been able to achieve with wildtype SpCas9, and this is what the title of the manuscript indicates. That we included what turned out to be negative controls in some experiments strikes us as unlikely to mislead readers.

Despite these major issues that prevent this reviewer from positively moving forward with this manuscript, several additional concerns exist:

1) The wide range of correlations (0.25 to 0.87) for Cas9 variants during PAM mapping is highly concerning. Regardless of the activity of an enzyme, this reviewer would expect that replicates should overall correlate well. As the authors do not provide reasoning for the nature of this low reproducibility, this fact is concerning and questions downstream analyses and conclusions, at least from the low-reproducing screens.

Respectfully, we disagree with the reviewer's assertion that "regardless of the activity of an enzyme... replicates should overall correlate well."

As a thought experiment, consider cells that have no Cas9 in them at all: here, zero guides will have any activity, and thus changes in guide abundance are purely stochastic. If one were to conduct replicates of this experiment, the expectation is no correlation. It therefore follows that cells with low levels of Cas9 activity will have a low signal:noise ratio, and thus low correlation across replicates.

The one Cas9 variant that showed extremely low correlation (Cas9-VRER) indeed had very low activity, as can be seen in Figure 3b, where no PAM shows high activity. We have added text to the manuscript explaining this interpretation more thoroughly.

2) The authors computed precision recalls for essential/non-essential genes to interpret the PAM-mapping results of Cas9 variants. While this analysis is well established for comparing screen performance; at this location, however, this is an unfitting metric for comparing different Cas9 variants/screens, as the essential/non-essential genes were originally determined by WT-SpCas9 and not the other variants. In other words, different Cas9 variants may reveal a different set of essential/non-essential genes due to different biological requirements and characteristics. A fact that is also supported by the recent literature in which screening time as a sole changeable parameter contributed to vastly different sets of essential genes (Rahman et al., 2021).

We respectfully disagree with the reviewer's assertion. The essential / non-essential gold standard genes established by Traver Hart and colleagues in 2014 (PMID: 24987113) did not rely on CRISPR screens, but rather RNAi screens. That manuscript further validated these gene sets on the basis of evolutionary and expression criteria, which were particularly important for defining the truly non-essential genes. Only later did Hart and colleagues use CRISPR screens to validate their original list and add to it (PMID: 26627737), as they found that RNAi and CRISPR largely agreed on the set of core essential genes.

The reviewer raises the supposition that genes will show different essentiality profiles based on the nuclease that targets them. While we are not fully sure what the reviewer means by "different biological requirements and characteristics," fortunately, we have already conducted the experiment to rule out this potential confounder. Previously, we developed Cas12a for use in pooled screens (PMID: 32661438). In that manuscript, we compared a genome-wide Cas9 library (Brunello) to a genome-wide Cas12a library (Humagne). Figure 7, panel d (added here for convenience) shows that the viability phenotypes of genes are well-correlated across the two nucleases. Further, panel e shows that collapsing genes into KEGG gene sets (ribosome, spliceosome, etc.) further increases the correlation. These results strongly support the conclusion that the essentiality of a gene is not confounded by the nuclease used to target it.

Finally, we interpret the Rahman et al. manuscript (PMID: 34018332) differently, and do not agree with the reviewer that there are vastly different sets of essential genes. Rather, that study illustrates that the essentiality of mitochondrial genes is particularly time-sensitive. In the authors' own words (final sentence of the introduction):

These analyses highlight the prominence of mitochondria-related genes' dependency profiles in CRISPR screens, which we hypothesize is a result of protein stability and screen dynamics.

Likewise, in the first sentence of the discussion, the authors write:

Our analysis suggests a link between the strength of ETC-related [electron transport chain] gene dependency and the screen sampling time.

In sum, the authors of this study do *not* conclude that there are “vastly different sets” of essential genes based on screening time. Further, we note that the conclusions of the Rahman et al. study are consistent with the analysis conducted by the Broad and Sanger DepMap teams (PMID: 31862961), which compared screens conducted at the two centers for 21 and 14 days, respectively:

The Broad-exclusive enriched GO terms included classes related to mitochondrial and RNA processing gene categories and other gene categories previously characterized as late dependencies.

In sum, we believe that our use of a common set of essential genes across Cas variants is a valid approach.

3) The authors claim that WT-Cas9 does not show a preference for G19. However, when looking carefully into the associated figure, WT-Cas9 also shows a preference for G19 (although marginal), an effect that is likely masked by WT-Cas9’s overall high performance/precision recall.

We have modified the text to clarify that the preference for a G19 is marginal.

4) To assess the off-target activity of Cas9 variants, the authors did functional screenings in A375 cells. However, it is unclear if the authors used a reference genome sequence or the genomic sequence of the A375 cells to annotate and design perfect-matching gRNAs or gRNAs with single or double mismatched. Since mismatch tolerance is a very important characteristic of Cas9 variants that may even contribute to the PAM-mapping data, the author should clarify/discuss which strategy was used and reason why.

We used the reference genome. The consistency of genetic screens conducted across hundreds of cell lines in the DepMap shows that individual SNPs in cell lines do not have a large effect on screen performance. Of course, this does not mean that SNPs are *never* important, but rather is by far the exception rather than a common confounding factor in screen interpretation.

As to the effect of SNPs on the analyses conducted here, we note that SNPs are unlikely to confound off-target analysis, as we first require the putative perfect-match guide to show a strong phenotype, which is not likely to be possible if there is already a

mismatch due to a SNP. Thus any guides that do target a SNP are likely to be filtered out and thus not impact our analyses.

5) To assess the off-target activity of PAM-flexible Cas9 variants, the authors designed and screened a gRNA library containing perfect matched gRNAs, in addition to gRNAs with single and double mismatches. However, it is entirely unclear to this reviewer why only 50% of the perfectly-matching gRNAs were considered for downstream analyses. If only 50% performed well (argument of the authors for focusing on the 147 gRNAs), it raises the concern about the accuracy of the previously reported PAM-preferences for these variants. In line with this, the downstream result of good to perfect separation from single and double mismatches is thus not surprising, as the input data are highly biased towards performing gRNAs.

Filtering for the highest activity guides ensures that we have a wider dynamic range to assess loss or retention of activity based on single or especially double mismatches. In other words, a guide with a single mismatch is more likely to give rise to detectable activity if the corresponding perfect match guide is very active, such that we are more sensitive to off-targets by including this filter, not less. We have added text to the manuscript explaining this logic.

6) In the BRCA1 screen, the authors examined base editing activity by the separation of negative and positive controls, but fail to provide a quantitative value for screen/editing performance. This, however, has recently been done for knockout screens by computing the effect size (Cohen's *d*), a measure the authors should also consider to use.

Supplementary Figure 5 (now Supplementary Figure 6) does in fact provide a quantitative value for screen performance, the area under the ROC curve of true and false positives (i.e. ROC-AUC) which is widely-used in the field as an assessment of screen performance. We also quantitate editing performance during validation studies by sequencing the targeted locus.

7) The high false positive rates (3 out of 8 (37.5%), 2 out of 10 (20%)) in their BRCA1 and BCL2 validation experiments is concerning, and the authors should comment on this finding and discuss how this compares to WT-Cas9.

We observed a 0% false positive rate among guides in the *BCL2* validation experiments. The two guides that do not meet the "validated" criteria (Supplementary Fig. 7c, now Supplementary Fig. 8c) did not score in the primary screen and were only included in the validation experiments for consistency (i.e. each guide was validated in

parallel with ABE and CBE, even though it only scored with one base editor). As we wrote in lines 503-507 of the original submission:

Based on this criterion, 8 of 8 guide-BE combinations validated as true positives, and 2 of 2 validated as true negatives. Further, sg20, 21, and 22 validated with both ABE and CBE, whereas sg19 and sg23 validated with the base editor with which they scored in the primary screens, but not with the non-scoring base editor; indeed, these sgRNAs were predicted to make either a silent edit (sg23, CBE), or no edit (sg19, ABE).

In other words, we validated 2 of 2 instances of a *true negative*, and those instances should not be added to the denominator of a determination of false positive rates.

As for the higher false positive rate in the *BRCA1* validation screens, we saw a similar trend in validation rates when we screened for *BRCA1* and *BRCA2* mutations with WT-CBE, where 5/13 guides (38.5%) represented false positives from the primary tiling screen (Hanna et al, 2021), which we have now indicated in the text. This is in keeping with negative selection screens (or “down assays” in the parlance of Bill Kaelin, PMID 28524181) generally having more false positives.

8) In the *BCL2* validation, the authors reason that D103 mutations disrupts the alpha2 helix in the P4 pocket, a known and essential residue in Venetoclax binding. While this conclusion is intuitive, this entire part does not add to the presented result and should be moved to the conclusion/discussion section. In fact, each individual structural reasoning part (also for the other mutations) lacks experimental proof and should therefore be moved to the discussion. Moreover, the authors should clearly mark the presented structures with their associated PDB-ID references as well as “adapted structures (not experimentally confirmed)”.

We agree with the reviewer that indicating that the structures are adapted is important and have modified the manuscript accordingly.

As for where in the manuscript these analyses should be presented, we believe they fit narratively where they occur, but would certainly defer to the editor if there is agreement that it should be moved to the discussion.

Minor issues:

1) Inconsistent use of WTCas9 and WT-Cas9 in text and figures.

We have updated accordingly.

2) Fig 1e/f - why are only 5 variants shown?

Figures 1e and 1f focus on WT-Cas9 and the subset of variants classified as high fidelity variants (with the exception of HiFi Cas9, for reasons noted above). The PAM flexible variants appear in a later figure, as we believe that, narratively, it is logical to discuss them separately from enhanced-specificity from PAM-flexible variants, as they have separate performance criteria.

3) Fig 1d, 2b, 4b, SF4b, SF5d, - Y axis of density plots is missing.

All of the plots mentioned above (and in minor point 4) by the reviewer are stacked density plots. We find this to be a convenient way to show several sets of data in a condensed format, but since the plots are stacked, adding redundant y-axes renders them quite crowded. These y-axes would simply indicate the fraction of the population represented at each value on the x-axis - something that can be roughly intuited from the shape of the plot itself. We have found several examples of manuscripts with similar density plots that were recently published in Nature Communications (e.g. PMIDs: 32041945, 32286308, 34750397). Given this information, we believe that the addition of y-axes will subtract visually from the figures more than it will add any helpful information. If the editors disagree, we are happy to substitute these stacked plots with individual kde plots which would more easily accommodate y-axes.

4) Sup Fig 2 - 1) Typo: “work6”, 2) y-axis of kde plot missing.

Thank you for catching this typo, it has been fixed.

5) Line 205 - consistency of language: “showed intermediate activity” was “diminished” activity before.

We have used the reviewer’s preferred language.

6) Fig 3a - This appears to be a redundant graphic, i.e. what is the difference to Sup Fig 1a?

Figure 3a shows the activity of PAM-flexible variants at all 64 NGNN PAMs, while Sup Fig 1a (now Sup Fig 2a) shows the activity of 9 Cas9 variants (PAM-flexible and high-fidelity) at any PAM where at least one variant had a fraction active of ≥ 0.3 . We believe it serves to hone the focus of the reader in on the activity of the PAM-flexible

variants only, but do admit that it is somewhat redundant. If space becomes an issue, we can accommodate.

7) Sup Fig 3 text – “Kim at el31” reference is misspelled.

We have fixed this. Thank you.

Reviewer #2

The manuscript “Benchmarking of SpCas9 variants enables deeper base editor screens of BRCA1 and BCL2” by Annabel Sangree and colleagues describes the assessment of on- and off-target activities of SpCas9 variants based on their performance in pooled knock-out screens. From this benchmarking, the authors identify two variants with relaxed PAM requirements that are then used to demonstrate deeper tiling of endogenous genes in base editor screens.

General remarks.

An independent, side-by-side evaluation of SpCas9 variants as presented here is of interest to both the screening- and (to an extent) the precision editing communities. Base editor screens have shown great potential for a number of important applications. Their performance is proportional to their targeting resolution. The characterization of PAM-relaxed Cas9 versions in this context is thus important. Overall, the paper is comprehensive, the experiments and the results are well presented and the methods are well described. The conclusions are supported by the data and relevant literature is cited. The work is in my opinion suitable for publication in Nature Communications, both in terms of quality and impact.

We thank this reviewer for the nice summary and positive remarks.

Major points.

1) My only major concern is the comparability of the cell lines containing the SpCas9 variants. Different virus batches can have different titers, potentially resulting in different average copy-number of SpCas9 in the different lines, which would affect Cas9 protein levels. In addition, protein stability of the variants might differ. A fair benchmarking would require comparable expression levels of all variants. Thus, protein expression of

the different SpCas9 variants in the screened cell lines has to be assessed. If significant differences are found, the results need to be re-interpreted accordingly.

This is a good point. First, for all experiments already conducted, we were sure to transduce cells at low MOI to ensure that the vast majority of cells had only one integrant of Cas9 (which wasn't terribly difficult, as viral titers for large, Cas9-containing lentivirus are low).

Second, we have now added a new experiment, in which we freshly prepared lentivirus for all Cas9 variants and introduced them into A375 cells in parallel in order to assess Cas9 levels on a cell-by-cell basis via flow cytometry. Supplementary Figure 1a now shows that all variants are similarly expressed (included here for convenience). The SpG variant showed reduced expression via this assay, although it still performed well in screens compared to the other PAM-flexible variants.

Minor points.

1) The abstract states that 11 variants were assessed, Figure 1 has only 9 variants + WTCas9.

We agree that our initial phrasing was confusing, so we have rephrased that we evaluated 10 variants + WT Cas9 (rather than counting WT Cas9 itself as a 'variant').

The 10 variants include the 9 variants shown in Figure 1, plus HiFi Cas9. At the outset of this project, our initial clone of HiFi-Cas9 contained a mutation, which is why it was not included in Figure 1 (we have clarified this in the text) but we did assess it for off-target activity (Figure 2), bringing the total number of variants assessed to 10.

2) Throughout, it is not easy to follow which variants were assessed with which assays. A table summarizing this information would help. Please also explain why variants were included/excluded.

Indeed, this suggestion will help with minor point 1 as well. We have added a visual (Figure 1a) to help organize the manuscript, and have clarified our thinking behind why particular variants were selected for additional assays.

Reviewers' Comments:

Reviewer #1:

Remarks to the Author:

The reviewer wishes to thank the authors for carefully addressing the original concerns. In my view, the revised manuscript is substantially improved and my concerns have been fully answered by providing a detailed point-to-point response and adapting manuscript and figures.

The revised manuscript is well-suited for publication in Nature Communications.

Reviewer #2:

Remarks to the Author:

My comments have been satisfactorily addressed and I thus recommend publication of the work in Nature Communications.